



# Similarity and variability of blocked weather-regime dynamics in the Atlantic-European region

Franziska Teubler[1], Michael Riemer[1], Christopher Polster[1], Christian M. Grams[2], Seraphine Hauser[2], and Volkmar Wirth[1]

[1]Institute for Atmospheric Physics, Johannes Gutenberg-University Mainz, Mainz, Germany
[2]Institute of Meteorology and Climate Research (IMK-TRO), Department Troposphere Research, Karlsruhe Institute of Technology (KIT), Karlsruhe, Germany

**Abstract.** Weather regimes govern an important part of the sub-seasonal variability of the mid-latitude circulation. Due to their role in weather extremes and atmospheric predictability, regimes that feature a blocking anticyclone are of particular interest. This study investigates the dynamics of these "blocked" regimes in the North Atlantic-European region from a year-round perspective. For a comprehensive diagnostic, we combine wave activity concepts and a piecewise potential-vorticity
(PV) tendency framework. The latter essentially quantifies the well-established PV perspective of mid-latitude dynamics. All blocked regimes during the $1979 - 2021$ period of ERA5 reanalysis are considered.

Wave activity characteristics exhibit distinct differences between blocked regimes. After regime onset, one regime (Greenland Blocking) is associated with a suppression of wave activity flux, whereas two other regimes (Atlantic Ridge and European Blocking) are associated with a northward deflection of the flux without a clear net change. During onset, the envelope of
Rossby wave activity retracts upstream for Greenland Blocking, whereas the envelope extends downstream for Atlantic Ridge and European Blocking. The fourth regime (Scandinavian Blocking) exhibits intermediate wave activity characteristics. From the perspective of piecewise PV tendencies projected onto the respective regime pattern, the dynamics that govern regime onset exhibit a large degree of similarity: Linear Rossby wave dynamics and nonlinear eddy PV fluxes dominate and are of approximately equal relative importance, whereas baroclinic coupling and divergent amplification make minor contributions.
Most strikingly, all blocked regimes exhibit very similar (intra-regime) variability: a retrograde and an upstream pathway to regime onset. The retrograde pathway is dominated by nonlinear PV eddy fluxes, whereas the upstream pathway is dominated by linear Rossby wave dynamics. Importantly, there is a large degree of cancellation between the two pathways for some of the mechanisms before regime onset. The physical meaning of a regime-mean perspective before onset can thus be severely limited.

Implications of our results for understanding predictability of blocked regimes are discussed. We further discuss the limitations of projected tendencies in capturing the importance of moist processes, which tend to occur at the fringes or outside of the regime pattern. Finally, we stress that this study investigate the variability of the governing dynamics without prior empirical stratification of data by season or by type of regime transition. We demonstrate, however, that our dynamics-centered approach does *not* map predominantly on variability that is associated with these factors. The main modes of dynamical variability
revealed herein, and the large similarity of the blocked regimes in exhibiting this variability are thus significant results.



# 1 Introduction

The concept of weather regimes provides an important description of variability of the mid-latitude circulation on sub-seasonal timescales (Hannachi et al., 2017). Of particular importance are regimes that feature a blocking anticyclone, i.e., a quasi-stationary and long-lasting anticyclonic flow configuration that locally "blocks" the mean westerly flow and deviates the mid-

latitude jet. In these weather regimes, similar weather conditions occur in the same geographical region for a prolonged time, resulting in important sub-seasonal changes to local weather. Regions near the center of the blocking anticyclone may thereby experience temperature extremes whereas heavy precipitation and flooding may occur in adjacent regions (Rex, 1950; Kautz et al., 2022; White et al., 2022). Besides their importance for extremes, blocking anticyclones play an important but ambiguous role for atmospheric predictability. On the one hand, the longevity of blocking anticyclones implies a putative source of sub-

seasonal predictability (Buizza and Leutbecher, 2015). On the other hand, however, the correct representation of the life cycle of blocked weather regimes provides a major challenge to current numerical weather prediction models (Quinting and Vitart, 2019; Büeler et al., 2021), and the misrepresentation of the onset of blocking may lead to some of the largest forecast errors over Europe (Rodwell et al., 2013; Grams et al., 2018).

Generally, blocking exhibits large natural variability (Woollings et al., 2018). The goal of this study is to investigate vari-

ability and similarity in the dynamics of weather regimes that feature a blocking anticyclone specifically in the North Atlantic-European region (hereafter referred to as blocked regimes). Our study will employ a year-round classification of weather regimes (Grams et al., 2017). Several studies have demonstrated the significance of the weather regimes defined by this classification to describe variability of weather impacts and predictability on sub-seasonal timescale (e.g., Grams et al., 2017; Büeler et al., 2021). Blocked regimes constitute four out of seven regimes in this classification. The dynamics of blocked regimes are

linked to the formation and maintenance of blocking anticyclones, which have received decades of research interest. It should be noted, however, that the onset of a blocked regime does not necessarily imply the onset of blocking, because the onset of a blocked regime may be due to transition from another blocked regime. A number of different conceptual ideas have been developed to describe formation and maintenance mechanisms, which tend to emphasize different dominant mechanisms or flow features (e.g. Shutts, 1983; Benedict et al., 2004; Michel and Rivière, 2011; Yamazaki and Itoh, 2009; Pfahl et al., 2015;

Nakamura and Huang, 2018; Luo et al., 2019; Miller and Wang, 2022). The co-existence of this multitude of viable frameworks indicates that there are most likely different pathways to blocking, as has been demonstrated for subsamples of pathways (e.g., Nakamura et al., 1997; Drouard and Woollings, 2018).

The dynamics of weather regimes has often been investigated in terms of individual contributions to the evolution of the regimes' streamfunction patterns (e.g. Feldstein, 2002; Michel and Rivière, 2011; Luo et al., 2014; Miller and Wang, 2022).

This approach focuses on the evolution of upper-tropospheric vorticity and thus, essentially, implies a focus on dry, barotropic dynamics. However, the importance of cyclone activity (e.g. Lupo and Bosart, 1999), which may imply a role of baroclinic growth, and of moist processes (e.g. Tilly et al., 2008; Pfahl et al., 2015; Steinfeld et al., 2022) for the evolution of blocking anticyclones have been emphasized also. The potential vorticity (PV) perspective on mid-latitude dynamics (Hoskins et al., 1985) is able to capture both, the role of baroclinic interaction and the impact of moist processes on the upper-tropospheric





circulation (e.g. Davis et al., 1993; Pomroy and Thorpe, 2000; Chagnon et al., 2013; Teubler and Riemer, 2016, 2021; Riboldi
     et al., 2019; Spreitzer et al., 2019), in addition to quasi-barotropic dynamics. This study adopts the PV perspective and employs
     the quantitative, piecewise PV tendency diagnostic developed in the context of mid-latitude Rossby wave packets (Teubler and
     Riemer, 2021) and extended to the dynamics of a blocked regime in a case study (Hauser et al., 2022b). In short, the decom-
     position of dynamical mechanisms in this framework can be interpreted in terms of linear (quasi-barotropic) Rossby wave

dynamics, baroclinic interaction, divergent outflow associated with latent heat release below, direct diabatic PV modification,
     and nonlinear PV fluxes.

     This study considers all blocked regimes that occur in the $1979 - 2021$ period of the ERA5 reanalysis (Hersbach et al.,
     2020). A succinct description of the dynamics of the cases is thus required. Projecting piecewise tendencies that represent
     individual contributions to the governing dynamics onto a representative regime pattern provides such a succinct description

(e.g. Feldstein, 2002, 2003; Michel and Rivière, 2011). Essentially, these projections describe the contributions of individual
     mechanisms in strengthening or weakening the regime pattern. Tendencies projected onto the regime pattern, however, need
     to be distinguished from those that govern the evolution of the associated anomalies, in particular before the onset of the
     regime, when the spatial distribution of anomalies may differ substantially from that of the regime pattern (Feldstein, 2002).
     For example, projections do not account for processes that amplify PV anomalies outside of the regime pattern. This limitation

may affect in particular the diagnostic of the impact of the moist processes, which tend to occur upstream of the regime
     pattern (Neal et al., 2022; Hauser et al., 2022b). A distinct advantage of projections is, however, that variability in terms of
     mere geographical location of different patterns is eliminated and thus the variability of dynamical processes is more directly
     intercomparable. For this reason, and keeping the above limitation in mind, projections of piecewise PV tendencies onto regime
     patterns will be applied in this study as one means to examine variability and similarities between blocked regimes.

Piecewise PV tendencies provide information on local changes of PV. Inspection of spatial patterns of the tendencies enables
     interpretations in terms of wave dynamics. A more direct diagnostic of wave characteristics, however, is provided by the
     concept of wave activity and its flux. Improvements in diagnosing local, finite-amplitude wave activity (e.g. Nakamura and
     Huang, 2017; Ghinassi et al., 2018) help to apply these concepts to blocked regimes, which imply large-amplitude anomalies.
     Exploiting recent improvements, Nakamura and Huang (2017, 2018) have proposed a theory that likens the onset of blocking

to a traffic jam. The theory predicts that blocking onset occurs when incoming wave activity exceed the amount of wave activity
     that a jet is able to propagate downstream. In this model, before onset, blocking tends to be associated with an increased flux
     of wave activity upstream, and after onset with a decreased flux of wave activity downstream. PV and finite-amplitude wave
     activity are related concepts, and a decomposition of a wave activity budget equation into piecewise tendencies similar to that
     in our PV framework is feasible (Ghinassi et al., 2020). In this study, however, we will restrict ourselves to using the flux of

local finite-amplitude wave activity to describe variability of wave characteristics before and after the onset of blocked regimes.

     In the focus of many blocking models are interactions between different temporal (and spatial) scales (e.g. Shutts, 1983;
     Yamazaki and Itoh, 2009; Luo et al., 2014). Important aspects of these interactions are mediated by Rossby wave breaking
     (e.g., Benedict et al., 2004; Woollings et al., 2008; Michel and Rivière, 2011; Michel et al., 2021). A common approach
     is to decompose variables into different frequency bands, and to diagnose how nonlinear interactions between these bands





contribute to the low-frequency evolution of regimes. Arguably, this decomposition exhibits some degree of arbitrariness (Miller and Wang, 2022). The current study will evaluate a tendency equation for low-frequency PV anomalies that represent the evolution of blocked regimes, but we will refrain from investigating scale interactions in the nonlinear (PV eddy flux) term. Using the comprehensive framework of combined PV and wave activity diagnostics we will find already, without further frequency decomposition, distinct variability in the relative roles of linear and nonlinear dynamics, and of baroclinic and moist

contributions. An analysis of variability and similarity in the important aspect of scale interactions is thus deferred to future work.

It is well known that the occurrence frequency of individual weather regime and their characteristics vary with season (e.g., Cassou et al. (2005), supplementary material in Cassou (2008)). Weather regime dynamics are thus often studied with a focus on specific seasons (e.g., Cassou, 2008; Drouard and Woollings, 2018). Furthermore, there are preferred transitions between

weather regimes and a study of regime dynamics often focuses on these transitions (e.g., Evans and Black, 2003; Michel and Rivière, 2011). This study deviates from theses approaches in the sense that the variability in the dynamics of blocked regimes is studied without previous empirical stratification of the underlying data (by season or type of transition). In this sense, we give primacy to the dynamical mechanisms - as seen in our diagnostic framework - and investigate to what extent blocked regimes exhibit variability based on this dynamical information alone. This approach may be justified a priori by noting that

with the year-round definition the blocked regimes do indeed occur year-round: Scandinavian Blocking exhibits the largest relative seasonal preference, with a relative occurrence frequency of 6.5 % during core winter and 16.0 % during core summer (supplementary material in Grams et al., 2017). The other three blocked regimes are more evenly distributed. In the context of blocking, at least, a year-round perspective has been taken previously (Drouard et al., 2021). A posteriori, the approach is justified because we will find that the main modes of variability do *not* map predominantly on differences in season (e.g.,

extended summer vs. extended winter) or preferred regime transitions. Taking this approach in the current study does not imply that we believe that stratification of cases by season and regime transition is not of interest. In fact, we see this as a worthwhile future extension of the work presented herein.

Finally, we re-state the goal of this study: to provide a process-based, quantitative description of similarity and variability in the dynamical mechanisms that govern blocked regimes in the North Atlantic-European region. Our goal is not to test specific

proposed theories or conceptual models on a large number of real-atmospheric cases, although the interpretation of individual terms on our diagnostic framework is certainly motivated and informed by these theories and models.

The diagnostic framework employed in this study is introduced in section 2, along with the classification of weather regimes and the data we use. section 3 provides a composite-mean perspective on the individual blocked regimes, with a first discussion of similarity and variability. The main modes of intra-regime variability are investigated in section 4. A striking result here is

that this intra-regime variability is very similar between the blocked regimes. A summary and concluding discussion are given in section 5



## 2 Data and methods

### 2.1 Data

This study uses the European Centre for Medium-Range Weather Forecasts (ECMWF) re-analysis ERA5 data (Hersbach et al.,
2019) from 1979-2021 with a 3-hourly temporal resolution. We use a spatial resolution of $1°$ and 17 pressure levels (1000,
950, 925, 900, 850, 800, 700, 600, 500, 400, 300, 250, 200, 150, 100, 70, and 50 hPa), from which data is interpolated to a
50 hPa vertical resolution by cubic spline interpolation as input required for PV inversion. Subsequently, data is interpolated
onto eight isentropic surfaces (315-350 K, every 5 K) (interpolation scheme as implemented by May et al., 2022). To account
for seasonal variability, PV analysis is performed on isentropic levels that vary according to Röthlisberger et al. (2018) (320 K
in Dec, Jan, Feb, Mar, 325 K in Apr, Nov, 330 K in May, Oct, 335 K in Jun, Sep, and 340 K in Jul, Aug), and averaged
values within $\pm 5$ K around the varying central value will be used. For the computation of finite-amplitude local wave activity
in the QG framework, the re-analysis dataset is linearly interpolated to 41 equidistantly spaced levels of log-pressure height
$z = -H \ln(p/p_0)$ between 0 and 20 km, where $H = 7$ km and $p_0 = 1000$ hPa (interpolation scheme as implemented by Huang
et al., 2022).

### 2.2   Year-round definition of weather regimes

We use the year-round definition of seven weather regimes in the North Atlantic-European region (NAE; 80 °W-40 °E, 30 °-
90 °N) by Grams et al. (2017), adapted to ERA5. The definition of the regimes is based on geopotential height anomalies at
500 hPa calculated as deviations from a climatological background that is defined as the daily mean over the period 1979-
2019 and further smoothed by a 90-day running mean. Anomalies are filtered by a 10-day low-pass Lanczos filter (Duchon,
1979) to exclude high-frequency signals. After normalization of the anomalies for a year-round definition, k-means clustering is
performed for the expanded phase space of the seven leading empirical orthogonal functions (EOFs) that describe 74.4 % of the
variability. A weather regime is defined as the cluster mean of one of seven clusters, which was shown to be the optimal cluster
number in the year-round definition. The seven weather regimes consist of three regimes that are dominated by a cyclonic
anomaly (Zonal regime - ZO, Scandinavian Trough - ScTr, Atlantic Trough - AT) and four regimes that are dominated by an
anticyclonic anomaly (Atlantic Ridge - AR, European Blocking - EuBL, Scandinavian Blocking - ScBL, Greenland Blocking
- GL). The latter four regimes are the focus of this study.

To make a quantitative statement about the similarity of an instantaneous pattern to the seven weather regime patterns, we
use the weather regime index ($I_{WR}$) (Michel and Rivière, 2011; Grams et al., 2017) defined as

$$I_{WR}(t) = \frac{P_{WR}(t) - \overline{P_{WR}}}{\sqrt{\frac{1}{NT} \sum_{t=1}^{NT} [P_{WR}(t) - \overline{P_{WR}}]^2}}, \tag{1}$$

where NT is the total number of time steps within a climatological sample and $\overline{P_{WR}}$ the climatological mean of the projection

$$P_{WR}(t) = \frac{1}{\sum_{(\lambda,\varphi) \in NH} \cos \varphi} \sum_{(\lambda,\varphi) \in NH} \Phi^L(t,\lambda,\varphi) \Phi^L_{WR}(\lambda,\varphi) \cos \varphi, \tag{2}$$





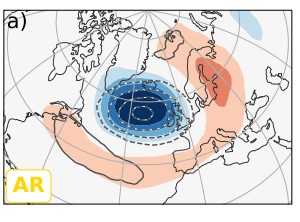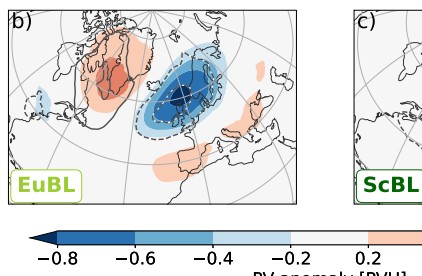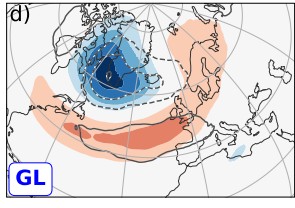

**Figure 1.** Your-round regime pattern in terms of PV on isentropic levels of blocked weather regimes: a) Atlantic Ridge (AR), b) European Blocking (EuBL), c) Scandinavian Blocking (ScBL), and d) Greenland Blocking (GL). Shading defines the PV regime pattern as composite of all days within a regime life cycle. Contours indicate the composite regime pattern at onset. The isentropic levels underlying these composites undergo seasonal variation as given in subsection 2.1.

with $\Phi^L(t, \lambda, \varphi)$ the low-frequency geopotential height anomaly at 500 hPa, $\varphi^L_{WR}$ the low-frequency geopotential height pattern that defines a weather regime, $\overline{P_{WR}}$ the climatological mean of the projection, and $(\lambda, \varphi)$ the respective longitude and latitude on the northern hemisphere (NH). Objective weather regime life cycles are derived based on the $I_{WR}$ for each regime and time step. Following Grams et al. (2017) a regime life cycle is defined as a persistent $I_{WR}$ above 1.0 for more than five consecutive days that shows for at least one time step the highest $I_{WR}$ of all seven regimes. A weather regime life cycle is called active, if $I_{WR} > 1.0$. The first time at that $I_{WR} > 1.0$ is defined as onset. A weather regime transition is defined as two subsequent active weather regime life cycles with less than 4 days in between. More detailed information on the criteria used in the definitions of the year-round regime life cycles can be found in Grams et al. (2017). The regime patterns in terms of PV, defined as the average of all days within the life cycle of the respective regime, are shown for reference for the four weather regimes dominated by an anticyclonic anomaly (a negative PV anomly in the northern hemisphere) in Figure 1. Generally speaking, regimes differ in the geographical location of the dominant negative anomaly as well as in the distribution of positive anomalies relative to that negative anomaly.

## 2.3 Diagnostic frameworks for mid-latitude dynamics

### 2.3.1 PV dynamics: piecewise-tendency framework for PV anomalies

We consider Ertel (1942) PV ($q$) on isentropic levels with the hydrostatic approximation: $q = \sigma^{-1}(\zeta_\theta + f)$, where $\zeta_\theta$ is the component of relative vorticity perpendicular to an isentropic surface, $f$ the Coriolis parameter and $\sigma = -g^{-1}(\partial p / \partial \theta)$ the isentropic layer density with gravity $g$, pressure $p$, and potential temperature $\theta$. The PV tendency equation is given by (adiabatic) advection along isentropes and nonconservative PV modification ($\mathcal{N}$):

$$\frac{\partial q}{\partial t} = -\boldsymbol{v} \cdot \boldsymbol{\nabla}_\theta q + \mathcal{N}, \tag{3}$$

with $(u, v, 0) =: \boldsymbol{v}$ the components of the horizontal wind and $\boldsymbol{\nabla}_\theta$ the gradient operator along an isentropic surface.





The basic idea here is i) to derive a tendency equation for the PV anomalies that are associated with the evolution of weather regimes, i.e., PV anomalies $q'_L$ that are subject to the same 10-day low-pass filter introduced in subsection 2.2; and ii) to decompose the advective PV tendency $\boldsymbol{v} \cdot \nabla_\theta q$ into individual terms that represent the PV perspective of mid-latitude dynamics (Hoskins et al., 1985; Davis et al., 1993; Teubler and Riemer, 2021). Piecewise PV inversion under non-linear balance (Charney, 1947; Davis and Emanuel, 1991; Davis, 1992) and a Helmholtz decomposition of the flow are employed to decompose the advecting wind field into the divergent flow and non-divergent components associated with upper- and lower-level PV anomalies. A detailed discussion of this decomposition technique is given in (Teubler and Riemer, 2021). Similar as in Hauser et al. (2022b), the tendency equation for $q'_L$ can symbolically be written as

$$\frac{\partial q'_L}{\partial t} \approx \text{WAVE}'_L + \text{ADV}'_L + \text{BC}'_L + \text{DIV}'_L + \text{EDDY}'_L . \tag{4}$$

We evaluate this equation on isentropes intersecting the mid-latitude tropopause, i.e., for upper-level PV anomalies. The isentropic levels thereby vary with season as given in subsection 2.1. The interpretation of the individual terms in Equation 4 is as follows. The terms WAVE and ADV describe the dynamics of linear, barotropic Rossby waves (Hoskins et al., 1985; Wirth et al., 2018; Teubler and Riemer, 2021). We refer to the sum of the two terms as quasi-barotropic dynamics (QB). The term WAVE represents intrinsic wave propagation with westward phase propagation (e.g., positive and negative tendencies straddling an existing positive PV anomaly upstream and downstream, respectively) and eastward intrinsic group propagation (e.g., positive and negative tendencies within an existing positive PV anomaly at the leading and trailing edge of a Rossby wave packet, respectively). The term ADV represents the advection of existing anomalies by the background flow [1]. The term BC describes baroclinic coupling with lower-level PV anomalies, which on average implies baroclinic growth. The term DIV represents the impact of the divergent flow. Large values of this term are usually associated with latent heat release below (see detailed discussion in Teubler and Riemer (2021); explicitly verified in a case study by Hauser et al. (2022b)) and can thus be interpreted as an indirect contribution by moist processes. The term EDDY describes the nonlinear redistribution of PV in terms of the convergence of the eddy flux of PV anomalies ($-\boldsymbol{\nabla} \cdot (\boldsymbol{v}'_{rot} q')$, where $\boldsymbol{v}'_{rot}$ is the non-divergent wind; hereafter eddy flux convergence for the sake of brevity).

Note that the subscript $L$ in Equation 4 denotes that the low-pass filter is applied to the tendency terms. The prime denotes that the deviation from the climatological average of the individual tendencies is considered. Because we here deviate in the computation of that climatological average from Hauser et al. (2022b), the derivation of Equation 4 is given in Appendix A, where we also explain why we do not consider nonconservative tendencies $N$ explicitly in this study.

### 2.3.2 Local finite-amplitude wave activity flux

Following the definitions and derivations of Nakamura and Huang (2018), let

$$q_g = \zeta_z + f \left[ 1 + e^{z/H} \frac{\partial}{\partial z} \left( \frac{e^{-z/H}(\theta - \tilde{\theta})}{\partial \tilde{\theta}/\partial z} \right) \right] \tag{5}$$

---

[1] sometimes referred to as Doppler shift





be quasi-geostrophic PV on the sphere with the vertical component of relative vorticity $\zeta_z$ and $\tilde{\theta}(z)$ the hemispheric-mean potential temperature. On every $z$ surface, we construct a so-called zonalized $Q_{\mathrm{g}}$ by rearranging $q_{\mathrm{g}}$ with an area-preserving procedure into a zonally symmetric state, ordered such that PV decreases monotonically from the North Pole to the Equator (Nakamura and Zhu, 2010). Local finite-amplitude wave activity $A$ at longitude $\lambda$ and latitude $\phi$ quantifies the meridional displacement of PV relative to this eddy-free zonalized state:

$$A(\lambda,\phi,z)\cos(\phi) = -a\int_0^{\Delta\phi} q_{\mathrm{g}}'(\lambda,\phi,z,\phi')\cos(\phi+\phi')\mathrm{d}\phi', \tag{6}$$

with eddy PV $q_{\mathrm{g}}'(\lambda,\phi,z,\phi') = q_{\mathrm{g}}(\lambda,\phi+\phi',z) - Q_g(\phi,z)$, the radius of Earth $a = 6378$ km and integral bounds from the latitude of evaluation ($\phi' = 0$) to the latitude of meridional displacement ($\Delta\phi$). Note that the domain of integration can be multi-segmented, e.g. in the presence of cut-offs (Huang and Nakamura, 2016). While LWA based on isentropic PV as used in the piecewise-tendency framework described above was constructed and applied by Ghinassi et al. (2018, 2020), we use LWA in the QG framework where the associated formalism is most advanced and has been used to study blocking previously (e.g. Nakamura and Huang, 2018; Neal et al., 2022).

On synoptic timescales, the column budget of local finite-amplitude wave activity is dominated by the convergence of the zonal flux of wave activity

$$F_\lambda = UA\cos(\phi) - a\int_0^{\Delta\phi} u'q_{\mathrm{g}}'\cos(\phi+\phi')\mathrm{d}\phi' + \frac{\cos(\phi)}{2}\left(v'^2 - u'^2 - \frac{R}{H}\frac{e^{-\kappa z/H}}{\partial\tilde{\theta}/\partial z}\theta'^2\right), \tag{7}$$

which is comprised of three terms: advection with the background state zonal wind ($U$, obtained from the zonalized atmosphere with no-slip boundary conditions for the PV inversion at the surface), the zonal component of the generalized Eliassen-Palm flux, and the Stokes drift, respectively (Huang and Nakamura, 2016, 2017). Eddy quantities $u'$, $v'$ and $\theta'$ in (7) are defined analogously to $q_{\mathrm{g}}'$, with $\phi' = 0$ held constant if they appear outside of an integral. In the following, we always consider the density-weighted column averages of $A\cos(\phi)$ and $F_\lambda$, temporally filtered with the low-frequency Lanczos filter introduced previously.

### 2.3.3 Envelope of Rossby waves

In addition to the local wave activity flux, we consider the envelope of Rossby waves as a complementary, phase-independent metric of the occurrence and amplitude of synoptic-scale waves. Following Zimin et al. (2006), we consider a zonally varying background and filter for wavenumbers 4-15. Instead of meridional wind anomalies, we here use wind anomalies perpendicular to the background flow for the envelope calculation. The background is defined by a 40-day low-pass filter. To account for the zonal asymmetry of troughs and ridges, the semi-geostrophic coordinate transformation by Wolf and Wirth (2015) is applied.

### 2.4 Leading modes of variability: EOF analysis and k-means clustering

To identify leading modes of variability, we calculate empirical orthogonal functions (EOFs) of spatial patterns with subsequent k-means clustering in principal-component space. The geographical region used for the EOF analysis is the same Atlantic-





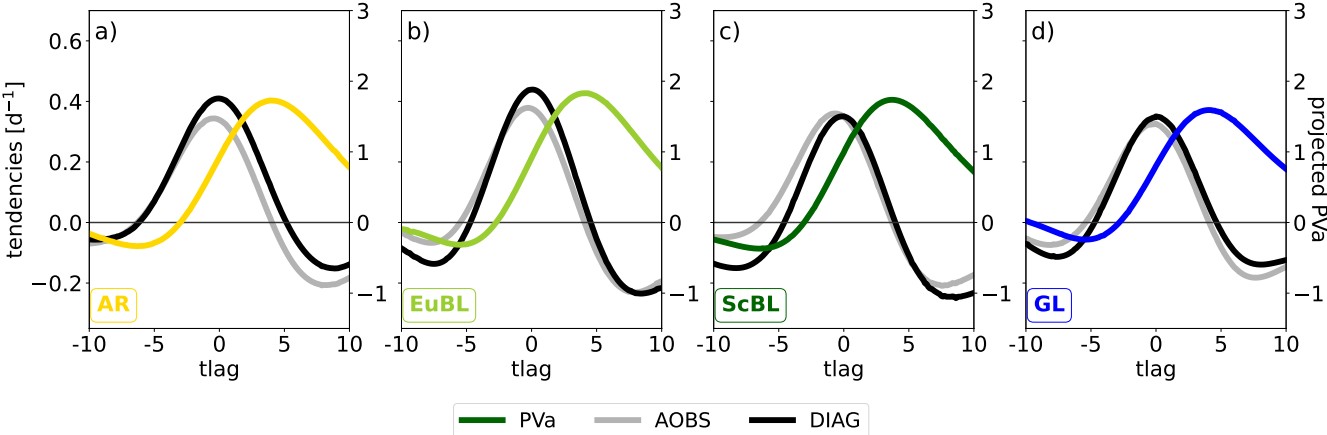

**Figure 2.** Projections onto the respective regime pattern of PV anomalies (colored, right y-axis), the associated observed tendency (grey, left y-axis), and the diagnosed tendency (black, left y-axis) shown ±10 days around onset: a) AR, b) EuBL, c) ScBL, and d) GL.

European region used for the definition of the weather regimes. We consider the first twelve EOFs, which describe at least 70% of the variability for all regimes and all considered underlying fields. For the k-means clustering, common heuristics to estimate the optimal cluster number, e.g., inspecting the described variance as a function of cluster number ("elbow plot"; Thorndike, 1953) or the ratio of intra- and inter-cluster variance (Caliński and Harabasz, 1974) do not yield unambiguous results. Because our interest here is on the leading-order variability of regime dynamics, we simply use two clusters.

### 2.5 Quantification of PV dynamics: Projection onto regime pattern

The relative contribution of low-frequency PV tendencies ($\partial q'^L/\partial t$) to the regime pattern is quantified by projecting the individual piecewise tendencies onto the regime pattern $q^L_{WR}$ (cf. subsection 2.2). This approach is similar to that by, e.g., Feldstein (2003) and Michel and Rivière (2011), who used projections of (inverted) vorticity tendencies to study the evolution of streamfunction patterns associated with the onset and transition of winter-time regimes. We here adapt this approach to PV dynamics. The projection is defined through

$$P_{\partial q^L/\partial t}(t)\big|_p := \frac{\sum_{(\lambda,\varphi)\in NH} \frac{\partial q^L(t,\lambda,\varphi)}{\partial t}\big|_j \, q^L_{WR}(\lambda,\varphi)\cos\varphi}{\sum_{(\lambda,\varphi)\in NH}(q^L_{WR})^2(\lambda,\varphi)\cos\varphi}, \qquad p \in \text{WAVE, ADV, BC, DIV, EDDY}. \tag{8}$$

The projection is thus defined as a normalized pattern correlation between the PV tendencies and the regime pattern. If the projection is positive, the respective process tends to build or maintain the regime pattern. If the projection is negative, the respective process weakens the pattern. To the extent that the weather regime index $I_{WR}$ (Equation 1) coincides with the projection of PV anomalies onto the regime pattern, the projected tendencies (Equation 8) quantify the contribution of individual processes to the evolution of the weather regime index. In the case study of EuBL, the close relation between the evolution of the PV-based and the geopotential-based regime index has been demonstrated (Hauser et al., 2022b, their Fig.4). It is important to note that the projection does not describe the contribution of processes to the evolution of instantaneous PV anomalies. These





processes may be distinctly different, in particular before onset, when the differences between the instantaneous PV pattern and the regime pattern can be expected to be large. For more details on the derivation and interpretation of the projection the reader is referred to (Feldstein, 2003, Sec. 5).

In Figure 2 the projected PV anomalies and the associated observed and diagnosed tendencies (the sum of the projections

of WAVE, ADV, BC, DIV, and EDDY) are shown for the different regimes ±10 days around regime onset. In general, the observed tendencies of the regime patterns are positive between day -5 and day 5 around onset. The evolution of all regimes is largely similar. The projected diagnosed tendencies describe this general evolution very well. The diagnosed tendencies tend to underestimate the observed tendencies before regime onset in particular for EuBL and ScBL, and tend to overestimate the tendencies thereafter, in particular for AR and except for ScBL after day 5. In general, however, diagnosed and observed

tendencies agree well, which warrants a more detailed analysis of the individual diagnosed contributions.

## 3 Mean perspective: Spatial composites and regime pattern dynamics

This section discusses differences between and similarities of the four blocked regimes from the regime-mean perspective. The next section demonstrates that the evolution leading to regime onset can be decomposed into two distinct pathways for all four regimes. Importantly, in terms of PV anomalies and piecewise PV tendencies, there is a large degree of cancellation between

these two pathways. This cancellation occurs because PV anomalies and PV tendencies tend to exhibit dipole patterns of positive and negative values and these patterns are approximately of opposite phase in the two pathways. A mean perspective of spatial patterns of PV anomalies and PV tendencies *before* onset is thus not informative of the mean evolution and will not be pursued. *After* onset, the PV patterns of the two pathways become - by definition of the regimes by spatial similarity - sufficiently similar for a mean PV perspective to be meaningful. The lower panels in Figure 3 illustrate the similarity of

the mean PV anomalies and the regime pattern after onset. Mean aspects of the Rossby wave envelope and the flux of local finite-amplitude wave activity (hereafter, for brevity: wave activity flux) are discussed before onset (subsection 3.1), because these metrics are by design largely phase independent and thus hardly suffer from cancellation. It turns out that relatively minor cancellation occurs also for the PV tendencies *projected* onto the regime pattern. The mean picture of the regime pattern dynamics are thus also considered before onset (subsection 3.3).

### 280 3.1 Blocked regimes and wave characteristics

This subsection demonstrates substantial differences between regimes in terms of synoptic-scale wave characteristics as seen by two complementary diagnostics: the Rossby wave envelope and the wave activity flux. With the Rossby wave envelope, we find spatial overlap of the envelope with the anticyclonic regime anomaly for AR and EuBL (Figure 3 e,f). In contrast, the anticyclonic regime anomaly is situated polewards and rather downstream of the envelope for ScBL and GL (Figure 3 g,h).

A difference between ScBL and GL is that the envelope overlaps with the cyclonic regime anomaly for ScBL, whereas the envelope ends upstream of the maximum of the cyclonic anomaly for GL. A comparison with the envelope before onset





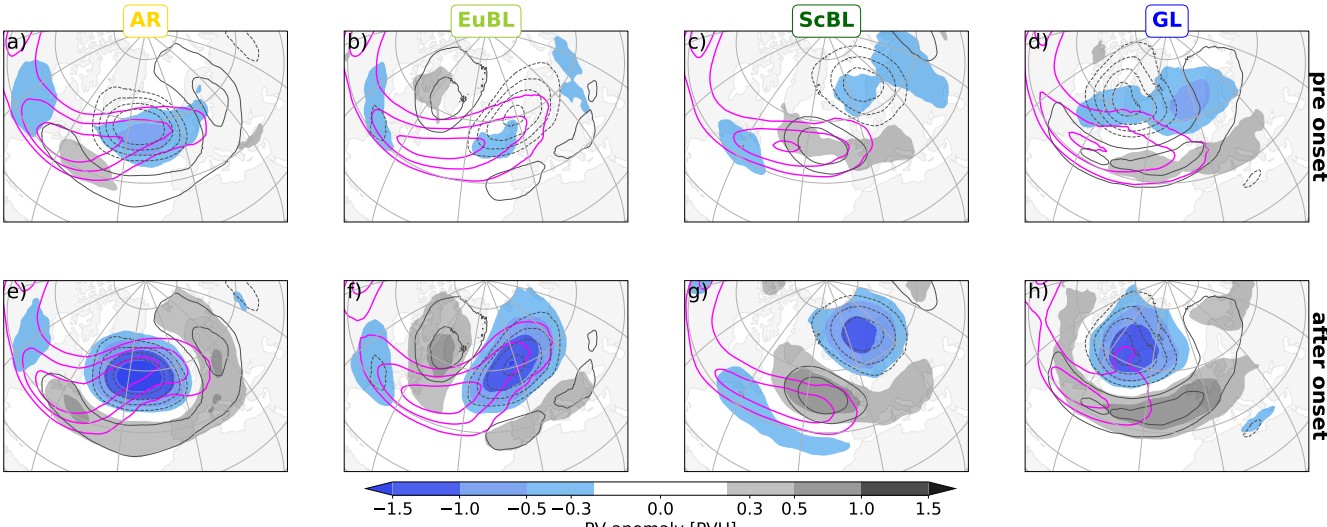

**Figure 3.** Composite maps of low-frequency PV anomalies (shading). Upper row (a-d): before onset (averaged from 3 days to 1 day before onset). Bottom row (e-h): after onset (averaged from $1-3$ days after onset). The respective regime is given at the top of each column. Magenta contours depict the envelope of synoptic-scale Rossby waves (for [16,18,20] m/s). Grey contours depict the PV regime pattern (for $\pm[0.2,0.4,0.6,0.8]$ PVU, negative values dashed).

(Figure 3 a-d) reveals that the envelope retracts (upstream) during regime onset for GL, and to lesser extent for ScBL. In contrast, the envelope extends downstream during onset for AR and EuBL.

Madonna et al. (2017) demonstrated a connection between the meridional jet locations and certain regimes: AR is associated
with a northern jet location, EuBL and ScBL with a southern branch in the Western Atlantic and a northern branch in the Eastern Atlantic, and GL with a southern jet location. These jet location coincide with our envelope metric after onset (Figure 3 e-h), except for ScBL, for which a northern jet branch over Europe is not indicated by the envelope. A possible explanation for this difference is that the northern jet branch does not constitute a continuous extension of the North Atlantic waveguide, along which Rossby wave packets would propagate, but rather constitutes a more local maximum of the zonal winds.

The wave activity flux as a complementary metric supports the notion of distinct differences between regimes. The prominent retraction of the Rossby wave envelope for GL is reflected in a major suppression of eastward wave activity flux in the North Atlantic (Figure 4 h). This pattern is consistent with the one-dimensional "traffic jam" model for blocking by Nakamura and Huang (2017, 2018), in which the onset of blocking effectively suppresses the zonal propagation of wave activity. A dipole of wave activity flux anomalies is evident for AR and EuBL, with enhanced and suppressed flux poleward and equatorward of the
anticyclonic regime anomaly, respectively (Figure 4 e,f). This dipole pattern signifies a deflection, rather than a suppression of wave activity transport, consistent with the downstream extension of the Rossby wave envelope during onset. The dipole pattern for ScBL is dominated by suppression of wave activity flux (Figure 4 g), which is again consistent with the retraction of





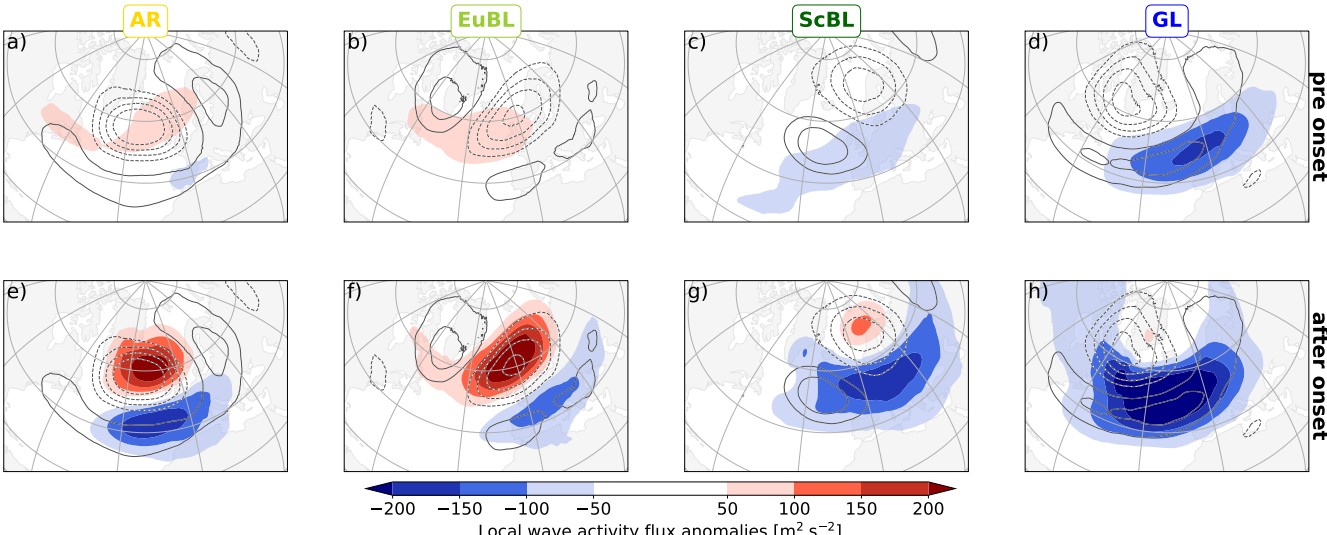

**Figure 4.** Same as Figure 3, but for low-frequency anomalies of the zonal local finite-amplitude wave activity flux $F_\lambda$ (shading).

the Rossby wave envelope. Our analysis thus suggests the interpretation that AR and EuBL occur within synoptic-scale wave packets (as suggested by Wang and Kuang, 2019) whereas GL does not. The interpretation for ScBL is less clear.

The "traffic jam" theory by Nakamura and Huang (2017, 2018) predicts that blocking onset is associated with enhanced upstream wave activity; more precisely: with wave activity that exceeds, in the region of the incipient block, the capacity of the "waveguide" to propagate wave activity downstream. A lower than usual "waveguide" capacity, however, may also favor blocking onset without enhanced upstream wave activity. Some indication of enhanced upstream wave activity flux before onset is found for AR and EuBL, but not for ScBL and GL (Figure 4 a-d). The enhanced synoptic-scale activity may be indicative of

the demonstrated importance of transient eddies for European blocking (e.g., Nakamura and Wallace, 1993; Evans and Black, 2003; Miller and Wang, 2022). As mentioned in the introduction, however, it should be kept in mind that the onset of blocked regimes here do not necessarily imply onset of blocking. We will further discuss these ideas in the context of different pathways to regime onset in section 4.

### 3.2 Spatial patterns of piecewise PV tendencies

Before presenting a very succinct depiction of the regime pattern dynamics, i.e., projections of piecewise PV tendencies onto the regime pattern, it is helpful to illustrate the spatial pattern of the PV tendencies. In addition, the spatial patterns themselves reveal similarities and distinct differences between regimes.

The tendencies due to linear, quasi-barotropic dynamics, i.e., tendencies due to intrinsic propagation and due to advection by the background flow (Figure 5a,d) exhibit a strong relation to the PV anomalies (cf. Figure 3). The pattern of the tendencies

associated with intrinsic propagation can be explained by cyclonic and anticyclonic circulations associated with positive and negative PV anomalies, respectively, and the resulting PV advection associated with a background PV gradient that is largely





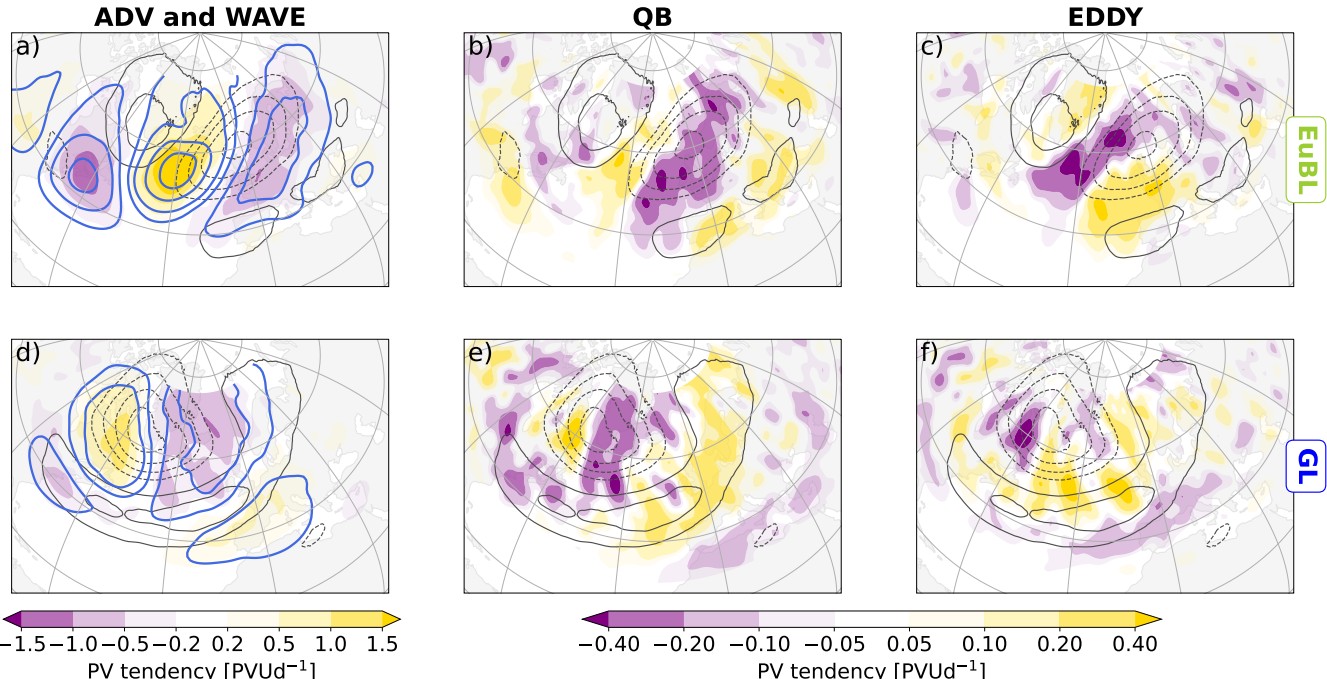

**Figure 5.** PV tendencies averaged over $1-3$ days after onset smoothed by a 1-2-1 smoother for visual clarity. Left column: Advection by the background flow (shading) and intrinsic wave propagation (blue contours, same values as color bar, negative values dashed). Middle column: Quasi-barotropic dynamics, i.e., the sum of the tendencies in the left column. Note the different color bar. Right column: Convergence of the PV eddy flux. Upper row for EuBL, lower row for GL. Black contours depict the respective regime PV pattern, every 0.2 PVU, negative values dashed, zero line omitted.

directed from south to north. The pattern of the tendencies associated with the advection by the background flow can be explained by the advection of existing PV anomalies by a largely zonal background flow. Importantly, both patterns resemble a wave train for EuBL, AR, and ScBL (exemplified for EuBL in Figure 5a), whereas for GL (Figure 5d) the pattern is largely

associated with the regime pattern itself, and in this sense much more local. Our interpretation of these differences is that EuBL, AR, and ScBL are associated with a larger-scale, low-frequency wave packet, whereas GL is not.

The tendencies due to intrinsic propagation and those due to advection by the background flow are approx. 180° out of phase, i.e., there is a large degree of cancellation between these two tendencies. Their net impact, i.e., the sum of the two tendencies is thus much smaller (in absolute values) than their individual contributions (Figure 5b,e, note the different color bar). For all

four regimes, positive (net) tendencies prevail in the cyclonic part of the regime pattern and negative (net) tendencies prevail in the anticyclonic part (illustrated for EuBL and GL in Figure 5b,e). For all regimes, linear quasi-barotropic dynamics thus amplify - on average - the respective regime pattern at this time (1-3 days after onset).

The tendencies due to eddy flux convergence exhibit a dipole of negative and positive values located poleward and equatorward of the negative regime anomaly, respectively (Figure 5c,f). A minor difference between regimes is that tendencies for





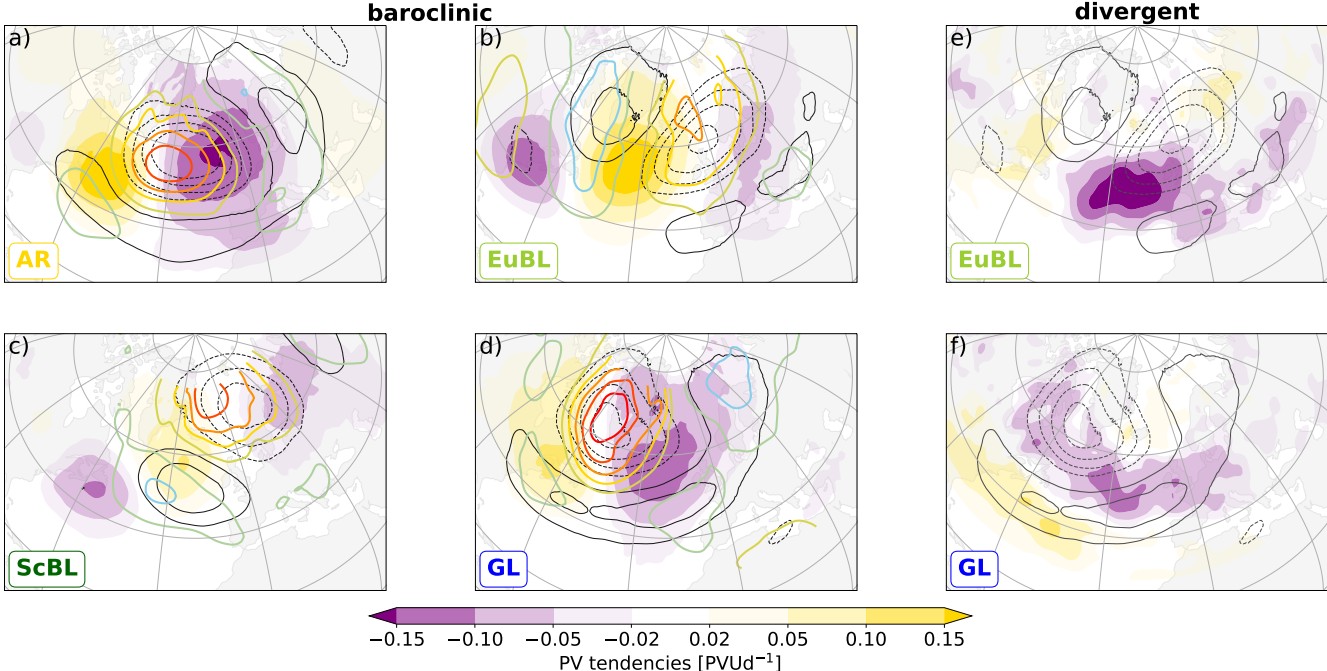

**Figure 6.** Same as Figure 5, but for baroclinic coupling for AR (a), EuBL (b), ScBL (c), and GL (d) and the divergent tendencies (EuBL (e) and GL (f)). Note the different color bar compared to Figure 5. Colored contours in a)-d) denote potential temperature at 850 hPa, warm colors for positive and cold colors for negative values, with a contour interval of 1 K, omitting the zero line.

EuBL and AR are spatially more coherent than for GL and ScBL (cf. Figure 5c,f for illustration). The dipole pattern reduces locally the positive poleward PV gradient and thereby decelerates the mid-latitude westerly flow, a key signature of blocking anticyclones (e.g. Illari, 1984). In this sense, the nonlinear dynamics of all regimes are similar at this time.

Systematic differences between regimes exist in terms of baroclinic coupling (Figure 6 a-d). The strongest low-level warm anomalies occur for AR and GL, located underneath the negative PV anomaly of the regime pattern (Figure 6 a,d). Accordingly, a dipole of relatively large positive and negative baroclinic tendencies straddles the negative anomaly. For GL, however, the vertical structure is approximately barotropic, whereas there is a small but notable upstream shift of the warm anomaly relative to the upper-level anomaly for AR. For EuBL and ScBL, the warm anomalies are weaker but more prominently shifted upstream (Figure 6 b,c). Moderate cold anomalies are found upstream of the positive PV anomalies of both regime patterns. Overall, EuBL and ScBL thus exhibit a rather baroclinic structure compared to AR and GL. Associated baroclinic tendencies, however, are either relatively weak (ScBL) or are predominantly located equatorward of and, due to meridional tilt, out of phase with the regime pattern anomalies (EuBL).

The divergent tendencies for EuBL, AR, and ScBL exhibit a distinct minimum just upstream and equatorwards of the negative regime anomaly (illustrated for EuBL in Figure 6 e). This pattern is consistent with that in the case study by Hauser et al. (2022b). In that case, the divergent tendencies projected little on the amplification of the regime pattern but, outside of





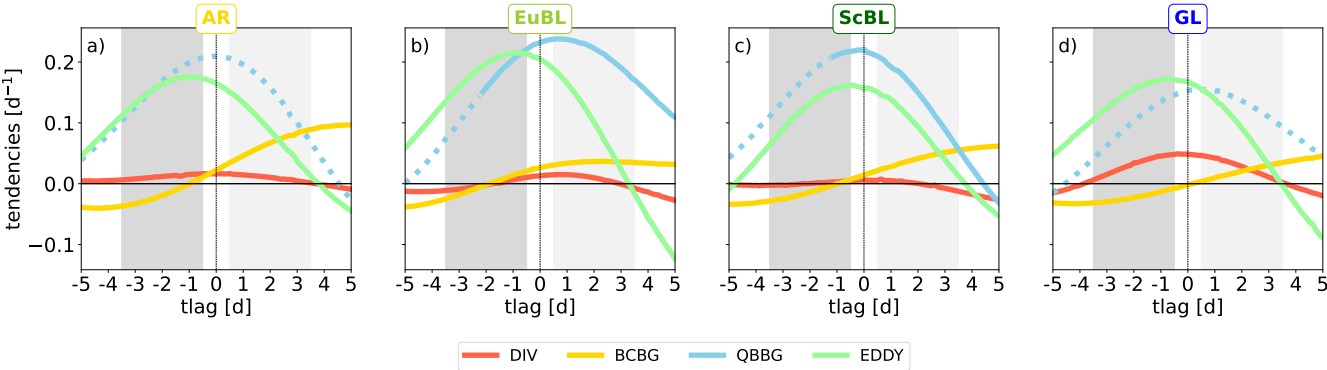

**Figure 7.** Piecewise PV tendencies projected onto the respective regime pattern for $\pm 5$ days around onset: a) AR, b) EuBL, c) ScBL, and d) GL. Blue: Quasi-barotropic dynamics, dotted if intrinsic propagation (WAVE) dominates advection by the background flow (ADV), solid otherwise. Yellow: Baroclinic coupling. Red: Divergent tendency. Green: Eddy flux convergence. Grey shading indicates the time periods that define the before-onset averages (dark) and after-onset averages (light) discussed as spatial maps, e.g., in Figure 3-Figure 6.

the regime pattern, contributed crucially to the amplification of the negative PV anomaly that later developed into the blocking anticyclone. Divergent tendencies for GL are weaker, less localized, and with a minimum downstream (instead of upstream) of the negative regime anomaly. The spatial pattern of the divergent tendencies are thus qualitatively similar for EuBL, AR, and ScBL, but distinct for GL.

### 3.3 Mean perspective of regime pattern dynamics: projections

The regime definition is based on EOF analysis of spatial patterns. By definition, different regimes thus differ in the geographical location of their respective anomalies (Figure 1). Apparently, this difference translates to differences in the geographical distribution of the associated dynamical mechanisms (Figure 5 and Figure 6). For a succinct comparison of the dynamics of different regimes, the impact of these geographical differences should be minimized. One way to do so is by projecting the tendencies associated with individual mechanisms onto the regime pattern (e.g., Feldstein, 2003; Michel and Rivière, 2011).

Projections of the PV tendencies are directly linked to the evolution of the weather regime index and thus give insight into the mechanisms contributing to the evolution of the different regimes (see subsection 2.5).

The individual contributions to the different regimes are shown in Figure 7 for $\pm 5$ days around regime onset. From this perspective, the dynamics of all four regimes are clearly dominated by linear quasi-barotropic dynamics and (nonlinear) eddy flux convergence, which in general increase in amplitude from -5 days to onset, decrease thereafter, and are of approximately

the same relative importance before onset. The relative importance of both, baroclinic coupling and the divergent flow are small before onset. Importantly, before onset, the dynamics of all four regimes exhibit a large degree of similarity.

After onset, differences between the regimes increase: i) EuBL is most clearly dominated by linear, quasi-barotropic dynamics (Figure 7b), ii) advection by the background flow dominate for EuBL and ScBL (Figure 7b,c), whereas intrinsic propagation





dominates for AR and GL (Figure 7a,d), and iii) AR exhibits a large baroclinc growth (Figure 7a). Note that the (relatively dis-
tinct) baroclinic structure of ScBL and EuBL observed above (Figure 6) does not lead to a distinct role of baroclinic coupling.

The divergent contribution is largest for GL. Figure 6 clearly illustrates that this difference is not due to the amplitude of the
divergent tendencies: GL exhibits smaller amplitude than EuBL (cf. Figure 6 c,f) but negative tendencies overlap prominently
with the negative PV anomaly of the regime pattern for GL. In contrast, for EuBL (representative for AR and ScBL), the
strong divergent tendencies are largely located just upstream of the negative PV anomaly of the regime pattern. The largest
contribution to the regime pattern dynamics in GL is thus solely due to the location of the divergent tendencies relative to the
regime pattern.

## 4 Variability within blocked regimes

The succinct description of regime dynamics in Figure 7 suggests that, on average, the dynamics of regime onset are very
similar for all four regimes. If distinct pathways to onset exist, however, the average picture would not be representative of
either pathway. For GL, e.g., a semi-Lagrangian perspective (Hauser et al., 2022b) reveals two distinct geographical origins of
the negative PV anomalies that later form the regimes' dominating anticyclonic anomaly[2] (Hauser et al., 2022a). In this section,
we thus explore the main modes of variability that underlie the average picture of regime onset. As suggested for GL, different
pathways to the same regime pattern may manifest themselves in differences in the spatial distribution of the associated PV
anomalies. Our investigation of intra-regime variability will thus be based on this spatial distribution before regime onset. For
each regime individually, we perform EOF analysis and subsequent k-means clustering, as detailed in subsection 2.4, on the
spatial pattern of PV anomalies averaged before onset (specifically: from day -3 to day -1).

### 4.1 Distinct modes of variability: Retrograde and upstream pathways

The cluster-mean PV anomalies before onset are shown for all four regimes in Figure 8. In terms of the location of a negative
PV anomaly relative to the regime pattern, the variability in all regimes is strikingly similar: One cluster features a negative PV
anomaly that is located downstream of the maximum of the negative anomaly of the respective regime pattern (Figure 8 a-d),
the other cluster features a negative PV anomaly that is located upstream (Figure 8 e-h). We henceforth refer to these clusters
as retrograde and upstream, respectively. For AR and GL, both clusters occur with similar frequency, whereas the retrograde
cluster occurs approx. 30 % more frequently for EuBL (103 vs. 78 cases) and approx. 40 % less frequently for ScBL (71 vs.
118 cases). The retrograde clusters of all regimes exhibit a downstream negative anomaly of high amplitude (Figure 8 a-d).
Visual inspection of the cluster-mean PV anomalies at individual times before onset (from day -3 to day 0; not shown) reveals
that this negative anomaly moves upstream, i.e., retrogrades towards the center of the regime pattern with time, with little
change in amplitude. The negative mean anomaly in the upstream clusters exhibits less amplitude than those in the retrograde

---

[2]Some indication of these distinct origins can be seen in the two local minima of PV anomalies before onset to the east and southwest of Greenland in
Figure 3 d





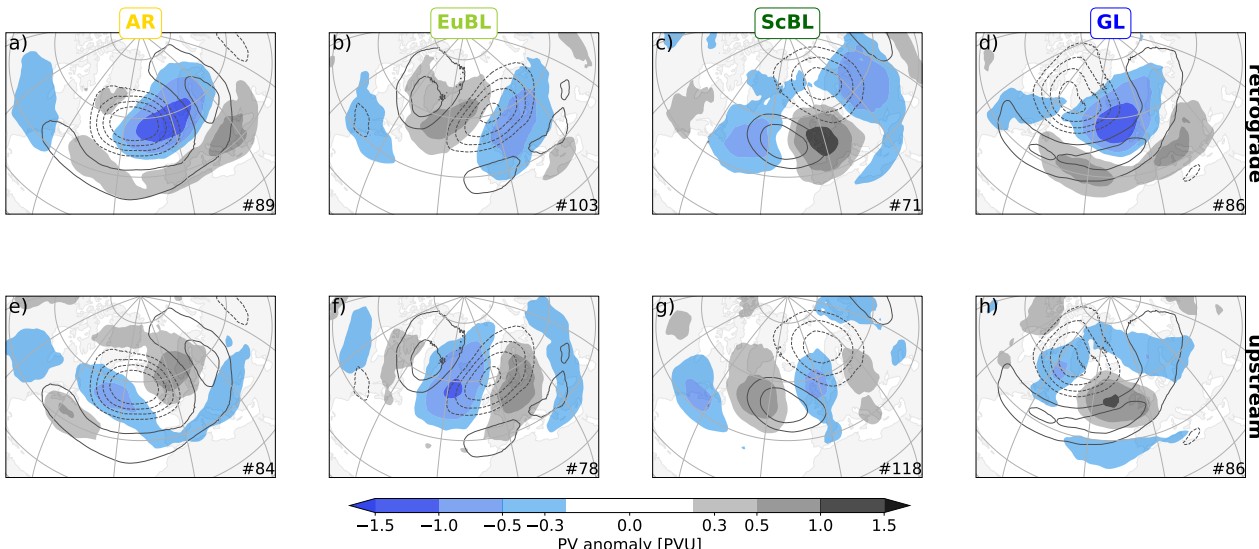

**Figure 8.** Same as Figure 3, but for the retrograde (top row) and upstream cluster (bottom row), averaged over $1 - 3$ days before onset.

clusters, except for EuBL (cf. Figure 8 b,f). The cluster-mean negative anomalies in the upstream clusters amplify while they move downstream towards the center of the regime pattern from day -3 to day 0 (not shown).

The PV anomaly patterns in the upstream clusters indicate that regime onset is associated with a larger-scale wave-like pattern, except for GL. The same is true for the retrograde clusters of EuBL and ScBL. For these two regimes, the wave-like patterns in the two respective clusters are largely $180°$ out of phase, i.e., there is a large degree of cancellation between positive and negative PV anomalies when considering the mean of all cases of these regimes (cf. Figure 3). Evidently, the mean perspective may largely conceal these important wave-like patterns.

**4.2   Nonlinear eddy fluxes vs. linear wave dynamics**

The dynamical mechanisms governing the formation of the regime pattern for the two clusters for all four regimes are shown in Figure 9. Evidently, for all regimes, the two different clusters exhibit very different dynamics. Importantly, these differences are much more pronounced than i) the differences between the mechanisms that govern the individual regimes in the mean sense (Figure 7) and ii) the differences between the individual regimes within the retrograde and the upstream clusters, respectively.

Linear, quasi-barotropic dynamics dominate the upstream clusters (Figure 9, bottom panels). These dynamics are directly linked to the PV anomalies (as discussed in 3.2). Nonlinear eddy fluxes dominate the retrograde cluster (Figure 9, top panels). These fluxes can be interpreted in terms of the self-advection of PV anomalies (Equation A7 - Equation A6), which is again directly linked to the PV anomalies by PV inversion. Differences in these two dominating mechanisms can thus be expected for distinct differences in the distribution of PV anomalies. The similarity of the dynamical mechanisms for different regimes

within the same (retrograde or upstream) cluster, however, is a nontrivial and striking result.





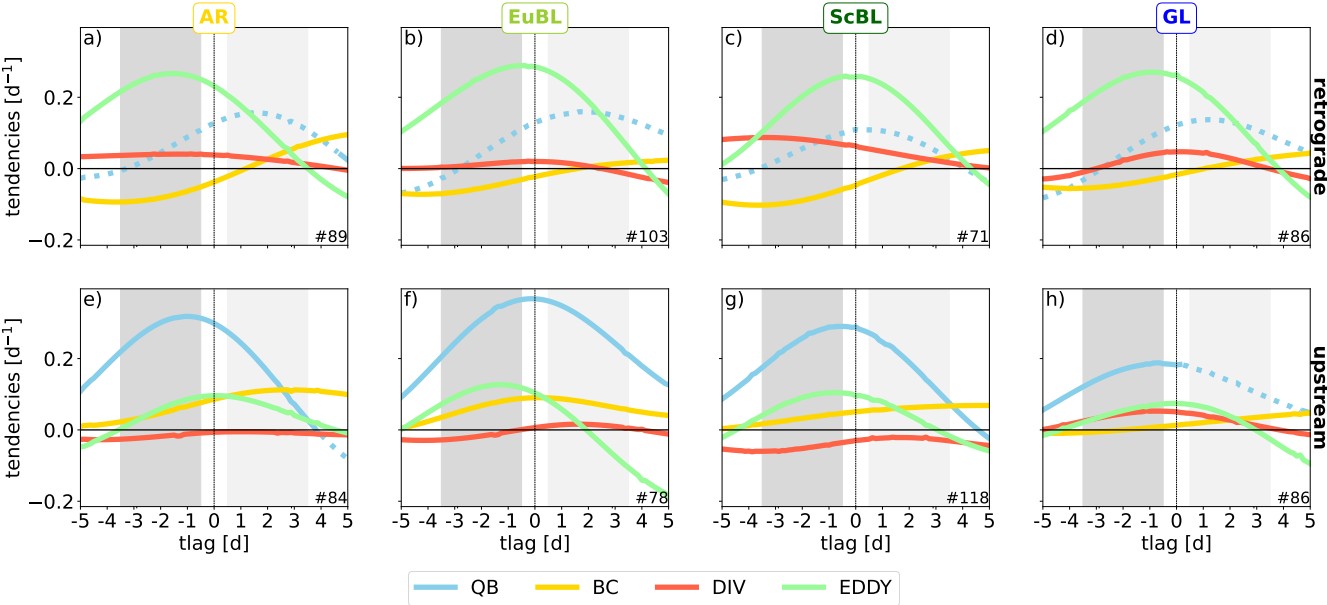

**Figure 9.** Same as Figure 7, but for the retrograde (top row) and the upstream cluster (bottom row).

In the retrograde cluster, the second-largest contribution, linear quasi-barotropic dynamics, increases in relative importance after onset and becomes dominant after day 2 (except for ScBL). For this contribution, intrinsic wave propagation dominates, as one may expect from linear theory for a retrograding Rossby wave. Note, however, that the observed retrogression, i.e., the upstream displacement of the PV anomalies leading to regime onsets is here dominated by nonlinear dynamics (i.e.,

eddy fluxes). Baroclinic coupling makes a consistently negative contribution before day 1, which is small (in absolute value) compared to the dominating nonlinear contribution, but comparable in amplitude to the linear quasi-barotropic contribution. The divergent contribution is relatively small and mostly positive. A minor inter-regime difference in the retrograde cluster is that the divergent contribution is notably larger for ScBL than for the other regimes.

    In the upstream cluster (Figure 9 e-h) the linear quasi-barotropic dynamics are largely dominated by the advection of PV

anomalies by the background flow. The dominance of this term indicates that important processes that amplify the advected anomalies, in particular moist processes, may occur outside of the regime pattern, and thus may only marginally be described by the tendencies projected onto the regime pattern (e.g., as described in the case study by Hauser et al., 2022b, from a semi-Lagrangian perspective). From the Eulerian perspective of the current study, however, we do not find a comparable signal: Negative divergent tendencies do not overlap spatially with the negative PV anomaly before onset (PV anomalies shown

in Figure 8; the spatial distribution of divergent tendencies is discussed in subsection 4.3). The lack of a signal could be associated with the temporal filtering here applied to both the PV anomalies and the PV tendencies, whereas Hauser et al. (2022b) consider the instantaneous distribution of these quantities. To test the generality of Hauser et al.'s results, it seems necessary to apply their semi-Lagrangian perspective to a large number of cases. Nonlinear dynamics and baroclinic coupling



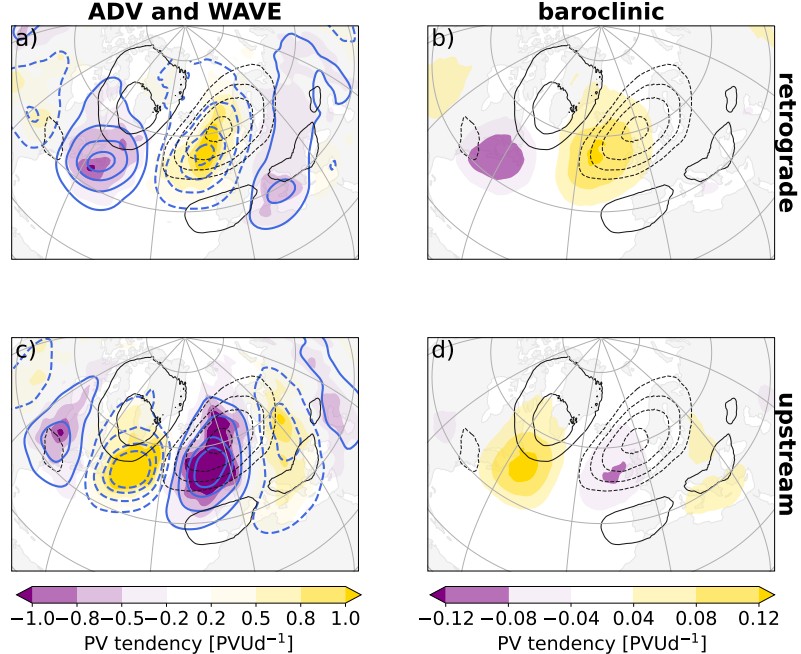

**Figure 10.** Same as Figure 6, but for the retrograde (upper row) and the upstream (bottom row) cluster for EuBL and averaged $1 - 3$ days *before* onset. Intrinsic wave propgation (blue contours) and advection by the background flow (shading) in the left column. Right column: baroclinic coupling. Note the different color bars

make further positive contributions to regime onset (before day 1), which are similar in amplitude, except for GL, for which
baroclinic coupling is very small but the divergent contribution is comparable to that of the nonlinear dynamics. The divergent
contribution is very small in the other three regimes. A further difference for GL is that the amplitude of both the linear and
nonlinear quasi-barotropic dynamics is 30-50 % smaller than in the other regimes.

### 4.3 Similarity and variability of spatial patterns

Two main characteristics of the differences between the retrograde and the upstream cluster are that before onset i) the linear
quasi-barotropic dynamics is dominated by intrinsic propagation in the retrograde cluster and by advection by the background
flow in the upstream cluster and ii) baroclinic coupling contributes negatively in the retrograde cluster and positively in the
upstream cluster (Figure 9). These characteristics can be explained by the spatial patterns of the associated tendencies, which
exhibit a phase shift of approx. $180°$ (Figure 10, exemplified for EuBL). The phase shift in the tendencies is tied to the
differences in the relative location of PV anomalies between the clusters (Figure 8). It is interesting to note that despite this
phase shift the (net) linear quasi-barotropic dynamics contribute positively in all clusters for all regimes (cf. Figure 9). For
the baroclinic coupling (Figure 10 b,d), which is less directly tied to the (upper-level) PV anomalies, positive and negative
contributions to the tendency pattern may have distinctly different amplitude for individual regimes in individual clusters (not





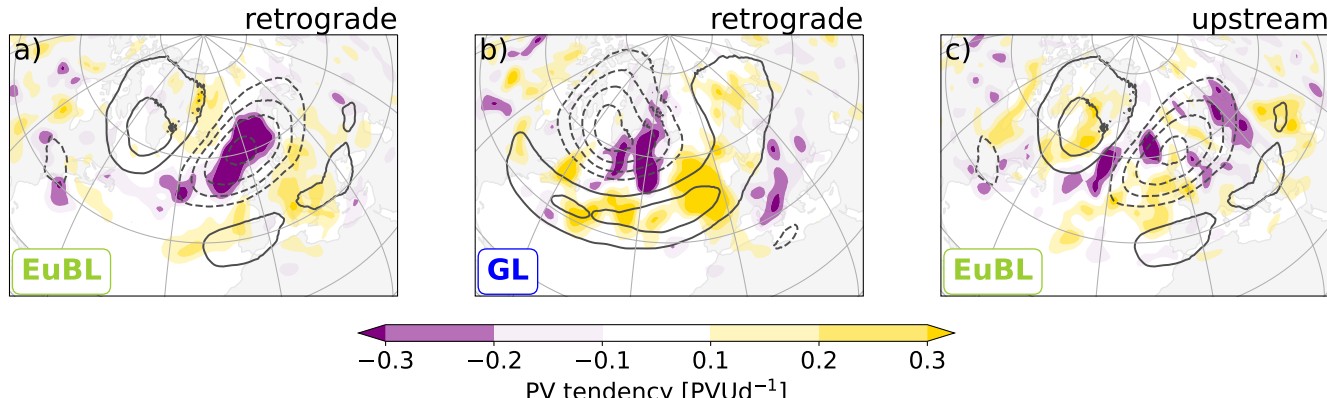

**Figure 11.** Same as Figure 6, but for the convergence of the eddy PV flux for the retrograde cluster for EuBL (a) and GL (b), and the upstream cluster for EuBL (c). Note the different color bar.

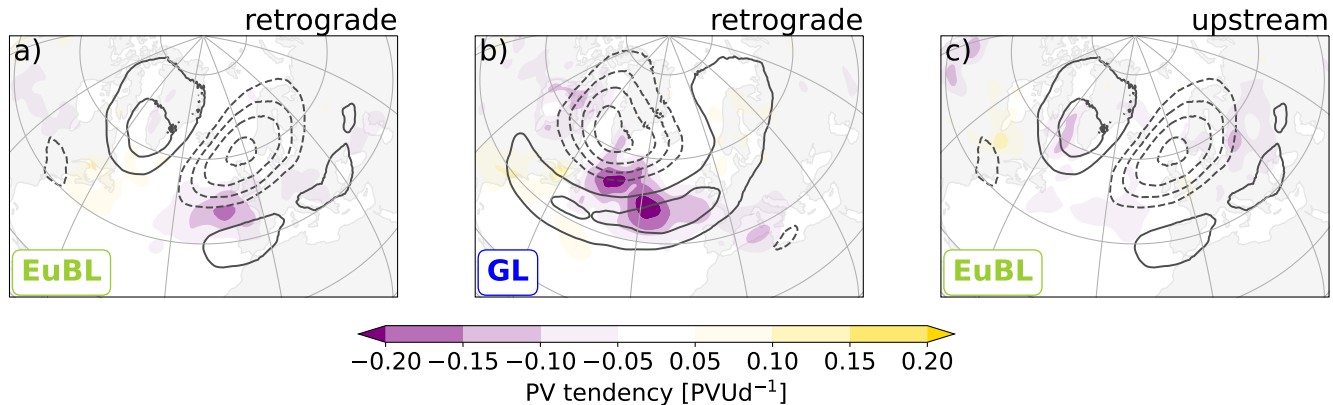

**Figure 12.** Same as Figure 6, but for the divergent tendency for the retrograde cluster for EuBL (a) and GL (b), and the upstream cluster for EuBL (c). Note the different color bar.

shown), but the phase shift between the retrograde and upstream cluster is a robust signal for all regimes. The nonlinear dynamics in the retrograde cluster are similar in all regimes in the sense that spatially coherent local maxima and minima occur within the cyclonic and anticyclonic regime anomalies, respectively (Figure 11a,b, exemplified for EuBL and GL, respectively). These extrema signify the nonlinear contribution to the regression of the associated PV anomalies during onset. For EuBL and AR, the local minimum dominates, whereas for ScBL the local maximum dominates (not shown). For GL, both extrema are of similar importance. In the upstream cluster, the spatial organization is much less clear (Figure 11c, exemplified for EuBL) and more variable between regimes (not shown).

The retrograde cluster exhibits similarity for the divergent tendency, in the sense that all regimes exhibit a prominent minimum associated with and equatorward of the regimes anticlone (Figure 12a,b, exemplified for EuBL and GL, respectively).





For EuBL (and AR, not shown) this minimum tends to be upstream of the anticyclone, and downstream for GL (and ScBL, not shown). In the upstream cluster, the spatial organization is again less clear and variability is larger between regimes. EuBL exhibits the least spatial organization (Figure 12 c). For AR, the pattern is similar to that in the retrograde cluster, for GL a
weak local minimum is located upstream and within the regime anticyclone, and for ScBL a prominent minimum is located within the regime cyclone (not shown).

## 5   Summary and discussion

### 5.1   Summary

We have investigated the dynamical mechanisms that govern weather regimes with a blocking anticyclone in the North Atlantic-
European region (blocked regimes) during the $1979 - 2021$ period of ERA5 reanalysis. A year-round perspective on weather regimes has been adopted (Grams et al., 2017). Our diagnostic framework comprises a piecewise PV tendency equation (Teubler and Riemer, 2021; Hauser et al., 2022b), which essentially quantifies the well-established PV perspective of mid-latitude dynamics (Hoskins et al., 1985), and projection of tendencies onto the regime patterns (e.g. Feldstein, 2002; Michel and Rivière, 2011). Advantages of projected tendencies are that they are directly related to the tendency of the respective
weather regime index, and that they eliminate differences in the mere geographical location of regimes. A major caveat of the projections is that they do not capture processes that occur outside of the regime pattern. Specifically, it has been demonstrated that this caveat limits the ability of the projections to capture amplification of negative PV anomalies, i.e., ridge amplification by divergent outflow prior to regime onset (Hauser et al., 2022b). We complement the projections by spatial (composite) maps of PV tendencies. We further complement the (local) piecewise PV perspective by diagnostics designed to more directly de-
scribe wave characteristics: the (synoptic-scale) Rossby wave envelope (Zimin et al., 2006; Wolf and Wirth, 2015) and local finite-amplitude wave activity fluxes (Nakamura and Huang, 2018).

Synoptic-scale Rossby wave characteristics exhibit distinct differences between the blocked regimes, most prominently between Greenland Blocking (GL) on the one hand and Atlantic Ridge (AR) and European Blocking (EuBL) on the other hand. After onset, GL is associated with a suppression of wave activity flux and the Rossby wave envelope retracts (upstream)
during onset. By contrast, AR and EuBL are associated with a northward deflection of wave activity flux without a clear net change. The Rossby wave envelope extends (downstream) during the onset of these regimes. Scandinavian Blocking (ScBL) exhibits intermediate characteristics: a northward deflection but with a net decrease of the wave activity flux, and a retraction of the Rossby wave envelope but to lesser extent than for GL. These results are largely consistent with the relation of the blocked regimes to the meridional jet position (Madonna et al., 2017), but the characteristics of the Rossby wave envelope and net
changes of wave activity flux provide new insights. The suppression of wave activity flux for GL is consistent with the "traffic jam" theory by Nakamura and Huang (2017, 2018). The deflection of wave activity flux found for the other blocked regimes, however, is a novel aspect that cannot be represented in their one-dimensional theory.

The governing dynamics of the blocked regimes, at least as seen in the projections of piecewise PV tendencies onto the respective regime pattern, exhibit a large degree of similarity. For all blocked regimes, i) linear, quasi-barotropic Rossby wave



dynamics and nonlinear eddy PV fluxes dominate and are of approximately equal relative importance, ii) baroclinic coupling contributes mostly negatively and is of small absolute magnitude, and iii) the divergent contribution tends to be positive but is also of small magnitude. We note that all blocked regimes exhibit a distinct minimum of divergent tendencies adjacent to the negative PV anomaly of the regime pattern, but mostly located outside of the regime pattern. The importance of these divergent tendencies is thus underrepresented by the projected tendencies. After regime onset, the differences in the governing dynamics

increase: The linear quasi-barotropic Rossby wave dynamics become increasingly dominant for EuBL (and to lesser extent for GL), whereas baroclinic growth becomes increasingly more important for AR and ScBL.

    Most strikingly, all blocked regimes exhibit very similar (intra-regime) variability before onset: a retrograde and an upstream cluster can be defined, in which the cluster-mean negative PV anomaly is located downstream and upstream of the negative PV anomaly of the regime pattern, respectively. The retrograde cluster is dominated by nonlinear dynamics (PV eddy fluxes),

whereas the upstream cluster is dominated by linear, quasi-barotropic (Rossby wave) dynamics. In the retrograde cluster, the baroclinic contribution is distinctly negative before onset, turning positive after onset. Inter-regime variability is found in the occurrence frequency of the retrograde and upstream cluster. Importantly, the spatial patterns of PV anomalies, the linear quasi-barotropic PV tendencies, and to lesser extent the baroclinic PV tendencies exhibit a large degree of cancellation between the two clusters before onset. A regime-mean investigation of these fields before onset is thus of little physical meaning.

## 5.2  Discussion

### 5.2.1  Regime transitions and seasonal dependence

This study investigates the variability in the dynamics of blocked regimes without prior empirical stratification by season or by type of regime transition. We here report briefly on the seasonal distribution of the two clusters (upstream and retrograde) and of their relation to regime transitions. In summer (JJA) the upstream cluster occurs approximately 35 % more frequently than

the retrograde cluster. This difference is almost exclusively attributable to a single regime: ScBL. For this regime the upstream cluster occurs about three times more frequently than the retrograde cluster. The upstream cluster for ScBL, however, occurs more frequently in winter (DJF) also, twice as frequently as the retrograde cluster. The retrograde cluster occurs overall 10 % more frequently in winter than the upstream cluster, mostly attributable to EuBL, in which the retrograde cluster occurs 60 % more frequently than in summer. Again, the retrograde cluster for EuBL occurs more frequently in summer also. Seasonal

variation alone can thus not be used as a proxy to describe the occurrence of the retrograde and the upstream cluster.

    With respect to regime transitions, the retrograde clusters for AR and GL show a preference for transitions from blocked regimes (in approximately 65 % of the cases), whereas the retrograde cluster of EuBL shows a clear preference for transitions from cyclonic regimes or no transitions (together 75 % of the cases). This result indicates that the retrograde cluster contains both, onset of blocking (for EuBL) and the (putative) displacement of an existing anticyclone during transition from another

blocked regime (for AR and GL). The retrograde cluster for ScBL does not exhibit preferred transitions. A signal for preferred transitions in the upstream cluster is less clear. Approximately 50 % of the transitions in the upstream clusters for EuBL and GL occur from another blocked regime. It is thus evident that transitions from another blocked regime populate both clusters.





The same is true for transitions from cyclonic regimes or no transitions. The different types of regime transitions thus do not provide a useful proxy for the two different clusters either. It is therefore confirmed that our dynamics-centered approach does
not merely reproduce variability that is associated with seasonal dependence or different types of regime transitions. Revealing the two main modes of dynamical variability, described by the retrograde and the upstream cluster, and the large similarity of the blocked regimes in exhibiting this variability is thus a significant result.

Furthermore, it is worth noting that wave activity flux anomalies upstream of blocked regimes *before* onset did not exhibit systematic differences between the upstream and the retrograde clusters and, through the lens of these clusters, no systematic
differences between preferred types of transitions. Although a clear signal does not emerge as a by-product of this study, clarifying the role of upstream wave activity fluxes for regime transitions is an important topic for future, more focused studies.

### 5.2.2 Variability of moist processes

Our investigation into the variability of blocked regime dynamics is based on the spatial distribution of PV anomalies before regime onset. This implies, as discussed in subsection 4.3, a direct link to the "dry" dynamics. Moist processes, however, may
be less constrained by the PV distribution. In addition, the divergent tendency, which we interpret as an indirect moist impact, has maximum amplitude outside of the regime pattern and is thus poorly represented by the projection of tendencies onto the regime pattern (Hauser et al., 2022b). The role of moist processes for main modes of dynamical variability may thus be underrepresented in this study. As a preliminary step towards mitigating this issue we have performed EOF analysis and k-means clustering on the spatial pattern of the divergent tendency, separately for each cluster of each regime. In the downstream
clusters of each regime new sub-clusters emerge, in which the difference between sub-clusters is largely in the amplitude of the divergent tendency. This amplitude signal may indicate a "moist" (large amplitude) vs. "dry" (low amplitude) mode of variability. Overall, the sub-clusters with weak divergent tendencies dominate, with an approximate occurrence frequency of 75% for GL, 60% for EuBL, 60% for ScBL, and 50% for AR. For the upstream cluster the results are inconclusive. We report this result here because we believe that the investigation of the variability of moist processes is a fruitful avenue for future work.
We stress, however, the limitations of an Eulerian approach to reliably capture this variability. Preferably, such an investigation would track the relevant negative PV anomalies and their amplification by the divergent flow with time, i.e., would employ a semi-Lagrangian approach as, e.g., in Hauser et al. (2022b).

### 5.2.3 Relation to predictability

One motivation for us to study the variability of dynamical mechanisms is to better understand the predictability of blocked
regimes. Higher predictability has been demonstrated for GL than for EuBL (Büeler et al., 2021; Hochman et al., 2021). From the perspective of dynamical mechanisms, one may expect that moist processes (here represented by the divergent tendency) and nonlinear processes (eddy PV fluxes) tend to exhibit lower predictability than linear wave dynamics. From the perspective of piecewise PV tendencies projected onto the regime pattern, however, GL is on average associated with a stronger nonlinear and divergent contribution than EuBL. In addition, the processes that govern the evolution of GL appear to be rather local,
whereas EuBL is embedded in a larger-scale wave pattern, which we would also rather expect to imply higher instead of




lower predictability. The only plausible explanation indicated in the projected tendencies may be associated with the larger role of baroclinc growth for EuBL (in particular in the upstream cluster). A plausible hypothesis could thus be that the lower predictability of EuBL is associated with a larger role of baroclinic, synoptic-scale activity, which may also comprise that part of the moist processes that are not captured by the projections onto the regime pattern.

A distinct difference between GL and EuBL is in the preferred types of transitions leading to regime onset (e.g., Büeler et al., 2021). Transitions from another blocked regime dominate for GL (57 % vs. 13 % from cyclonic regimes), whereas for EuBL transitions from a cyclonic regime occur with similar frequency (35 % vs. 33 % from another blocked regime). In other words, EuBL is more often than GL associated with blocking onset, for which it is known that forecast errors tend to be particularly large (Rodwell et al., 2013; Grams et al., 2018). Noting that error growth mechanisms may be distinct from the mechanisms

that govern the dynamics of the underlying flow (Baumgart et al., 2018; Craig et al., 2021), a further plausible hypothesis is that the observed difference in predictability is not primarily related to differences in the governing dynamical mechanisms but rather to differences in the flow-dependent error growth dynamics. In addition, Büeler et al. (2021) find a low-bias in representing the transitions from the zonal regime to EuBL (in the models considered in their study), i.e., a low bias in one pathway to blocking onset, which implies that model errors my further contribute to the observed differences in predictability. Certainly,

more future investigations are needed to substantiate the hypotheses put forth in this subsection.

    The piecewise PV perspective in combination with wave activity diagnostics provides a comprehensive quantitative framework to study the dynamics of weather regimes. Fruitful extensions of the current work include a decomposition of the eddy flux term to study interactions of different frequency bands, a focus on the dynamics of specific regime transitions and seasonal

differences, and a focus on the role of wave activity characteristics to different types of transitions into blocked regimes. It seems worth to mitigate the limitations of projected tendencies in future work, e.g., by employing the (more complex) semi-Langrangian approach of explicitly tracking those PV anomalies that eventually contribute to the formation or maintenance of a regime pattern. Finally, the relation between regime dynamics and regime predictability remains a further important topic for future research.

**Appendix A: Derivation of the low-pass filtered, piecewise PV tendency equation**

Let $< \cdot >$ denote the operator that defines the background state $q_0$ and associated $\boldsymbol{v}_0$. Here this operator is defined as daily averages of the years $1980 - 2019$ and a subsequent 30-day running mean. Anomalies $q'$ and $\boldsymbol{v}'$ are defined as deviations from the background state. It is

$$\frac{\partial q_0}{\partial t} := \frac{\partial < q >}{\partial t} = < \frac{\partial q}{\partial t} > = < -\boldsymbol{v} \cdot \boldsymbol{\nabla}_\theta q > + < \mathcal{N} >, \tag{A1}$$

where we used the PV equation (Equation 3). With $q = q_0 + q'$, the tendency equation for PV anomalies is

$$\frac{\partial q'}{\partial t} = \frac{\partial q}{\partial t} - \frac{\partial q_0}{\partial t} = -\boldsymbol{v} \cdot \boldsymbol{\nabla}_\theta q + \mathcal{N} - (- < -\boldsymbol{v} \cdot \boldsymbol{\nabla}_\theta q > + < \mathcal{N} >). \tag{A2}$$





Advective and nonconservative tendencies thus contribute to the evolution of PV anomalies to the extent that the tendencies differ from their "climatological mean" $< \cdot >$. Near the tropopause, where we evaluate the PV tendency equation, nonconservative tendencies ($\mathcal{N}$) are an order of magnitude smaller than advective tendencies, except for those associated with longwave radiation (Teubler and Riemer, 2021; Hauser et al., 2022b). Longwave radiative tendencies, however, have been shown to exhibit little coupling with other dynamical processes and can thus be considered to leading order as a "background process" (Teubler and Riemer, 2021). This background process $< \mathcal{N} >$ is here subtracted in the tendency equation for the anomalies (Equation A2). The remainder ($\mathcal{N} - < \mathcal{N} >$) is again small compared to the advective tendencies (not shown) and thus omitted from further analysis.

Let subscript $L$ denote a low-pass filter. We then get, now neglecting the nonconservative tendencies $\mathcal{N}$,

$$\frac{\partial q'_L}{\partial t} = \left(\frac{\partial q'}{\partial t}\right)_L \approx (-\boldsymbol{v} \cdot \boldsymbol{\nabla}_\theta q)_L - (< -\boldsymbol{v} \cdot \boldsymbol{\nabla}_\theta q >)_L . \tag{A3}$$

Making the further (very good) approximation that the climatological mean $< \cdot >$ can be treated as a constant with respect to the low-pass filter, we get

$$\frac{\partial q'_L}{\partial t} \approx (-\boldsymbol{v} \cdot \boldsymbol{\nabla}_\theta q)_L - < -\boldsymbol{v} \cdot \boldsymbol{\nabla}_\theta q >, \tag{A4}$$

which is essentially Equation 4 (where the deviation from the climatological mean is denoted by a prime and without the decomposition of the advective term).

The decomposition of the advective term proceeds as follows. Our decomposition of the wind field reads

$$\boldsymbol{v} = \boldsymbol{v}_0 + \boldsymbol{v}' = \boldsymbol{v}_0 + \boldsymbol{v}'_{div} + \boldsymbol{v}'_{rot} = \boldsymbol{v}_0 + \boldsymbol{v}'_{div} + \boldsymbol{v}'_{up} + \boldsymbol{v}'_{low} + \boldsymbol{v}'_{res} . \tag{A5}$$

The divergent[3] component $\boldsymbol{v}'_{div}$ is obtained by Helmholtz decomposition. The non-divergent component is denoted by $\boldsymbol{v}'_{rot}$. The wind components $\boldsymbol{v}'_{up}$ and $\boldsymbol{v}'_{low}$ are those associated with the upper- and lower-level PV anomalies[4], respectively, are obtained by piecewise PV inversion, which implies that they are non-divergent. The residual $\boldsymbol{v}'_{res}$ comprises any inaccuracies in the numerical methods and inherent uncertainties in piecewise PV inversion, i.e., non-linearity and imperfect knowledge of boundary conditions (discussed in detail in Teubler and Riemer, 2021). This residual does not affect the physical interpretation of our results and is henceforth omitted. With $q = q_0 + q'$ and $\boldsymbol{v} = \boldsymbol{v}_0 + \boldsymbol{v}'_{div} + \boldsymbol{v}'_{rot}$ we obtain six terms

$$\boldsymbol{v} \cdot \nabla_\theta q = \boldsymbol{v}_0 \cdot \nabla_\theta q_0 + \boldsymbol{v}'_{rot} \cdot \nabla_\theta q_0 + \boldsymbol{v}'_{div} \cdot \nabla_\theta q_0 + \boldsymbol{v}_0 \cdot \nabla_\theta q' + \boldsymbol{v}'_{rot} \cdot \nabla_\theta q' + \boldsymbol{v}'_{div} \cdot \nabla_\theta q' . \tag{A6}$$

For the first term, being a product of background variables, $\boldsymbol{v}_0 \cdot \nabla_\theta q_0 \approx < \boldsymbol{v}_0 \cdot \nabla_\theta q_0 >$. Moreover, the term is very small compared to all other terms to begin with because the background winds are approximately parallel to isolines of background PV. We thus neglect this term. In the next step, we combine the terms containing $\boldsymbol{v}'_{div}$, write $\boldsymbol{v}'_{rot} \cdot \nabla_\theta q'$ in flux form, use

---

[3]More precisely, the Helmholtz decomposition yields the irrotational wind. The harmonic component is negligible in our case and we may thus refer to the irrotational flow as divergent flow.

[4]Lower-level PV anomalies range from 850 hPa - 600 hPa and include the $\theta$ anomalies at the lower boundary (at 850 hPa. Upper-level PV anomalies are those at 650 hPa and above.





$\boldsymbol{v}'_{rot} \approx \boldsymbol{v}'_{up} + \boldsymbol{v}'_{up}$ in the term $\boldsymbol{v}'_{rot} \cdot \nabla_\theta q_0$ and re-order the resulting five terms:

$\qquad \boldsymbol{v} \cdot \nabla_\theta q \approx \boldsymbol{v}'_{up} \cdot \nabla_\theta q_0 + \boldsymbol{v}_0 \cdot \nabla_\theta q' + \boldsymbol{v}'_{low} \cdot \nabla_\theta q_0 + \boldsymbol{v}'_{div} \cdot \nabla_\theta q + \nabla_\theta \cdot (\boldsymbol{v}'_{rot} q') . \qquad\qquad\qquad$ (A7)

These terms finally correspond to (minus) the terms WAVE, ADV, BC, DIV, and EDDY in Equation 4, respectively.

*Code and data availability.* The data are referenced in Sect. 2.1. The codes and data from this study can be provided by the authors upon request.

*Author contributions.* FT calculated and provided the PV diagnostic. CP calculated and provided the local wave activity diagnostic. SH
helped to conceptualize distinct pathways and provided knowledge on weather regime characteristics. CMG provided the year-round North Atlantic-European weather regime data based on ERA5. FT, CP and MR wrote the manuscript together. CMG, MR and VW gave important guidance during the project and provided feedback on the manuscript.

*Competing interests.* MR and CMG are members of the editorial board of Weather and Climate Dynamics. The authors have no other competing interests to declare.

*Acknowledgements.* The research leading to these results has been done within the sub-project "Dynamics and predictability of blocked regimes in the Atlantic-European region (A8)" of the Transregional Collaborative Research Center SFB / TRR 165 "Waves to Weather" (www.wavestoweather.de) funded by the German Research Foundation (DFG). The contribution of CMG is funded by the Helmholtz Association as part of the Young Investigator Group "Sub-seasonal Predictability: Understanding the Role of Diabatic Outflow" (SPREADOUT, grant VH-NG-1243).





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
