# Peer review of "Similarity and variability of blocked weather-regime dynamics in the Atlantic-European region"

_Weather and Climate Dynamics, 2022_

## Referee Comment (RC2)

**Similarity and variability of blocked weather-regime dynamics in the Atlantic-European region**

Franziska Teubler, Michael Riemer, Christopher Polster, Christian M. Grams, Seraphine Hauser and Volkmar Wirth

This paper builds on previous work by the authors and others to examine how different categories of blocking are related to different kinds of forcing, derived from PV tendency analyses. It presents interesting new insight on the apparent similarity of the different categories of blocking, is clearly written and has good diagrams. My recommendation is to publish with minor corrections.

**Points for the authors to consider:**

l.76 'A distinct advantage of projections is, however, that variability in terms of mere geographical location of different patterns is eliminated and thus the variability of dynamical processes is more directly intercomparable.' How does this statement (repeated on l. 470) tally with equation 2, where the weather regime index is calculated by correlation of an instantaneous pattern to the weather regime patterns, the latter geographically fixed in terms of $\lambda$ and $\varphi$?

Fig 2. The most striking thing to me about fig 2 is that the four coloured traces are almost identical, yet the authors do not mention this in lines 259-265 where the figure is discussed. No doubt I have missed something very obvious in the previous discussion, but I can't be alone in this, so please could the discussion be extended to explain why the coloured curves are so similar?

L.355-361. This paragraph does not explain how projecting PV tendencies on to the regime pattern minimises the impact of geographical differences, and I can't see any obvious reason why it should. Explain!

**Typos and minor corrections**

l.59, 365 remove comma
l.85 exceeds
l.102 regimes
Fig 1 caption: Year-round
l.184 Brackets incorrectly applied around reference. Also similar to (not as)
l.203 nonconservative
l.215 define LWA (I presume it's the same as A, the local finite-amplitude wave activity)
l.258 Brackets incorrectly applied around reference.
l.292 locations
Fig 5 caption: explain that top row is EuBL and bottom row GL. The left hand panels confused me because (I think that) negative blue contours are shown as solid, same as the

positive ones (i.e. all contours are positive). According to my reading of the text l.327, some of them should be negative.

l. 380 'either' suggests that there are only two pathways, yet that is not how the text reads

l.456 regime's

---

## Author Comment (AC1)

**Response to Reviewers**

Franziska Teubler, Michael Riemer, Christopher Polster,
Christian M. Grams, Seraphine Hauser, and Volkmar Wirth

February 13, 2023

**Contents**

**1  Response to Gwendal Rivière**

The paper investigates processes leading to the formation of blocked weather regimes in the North Atlantic region. Among the 7 identified weather regimes based on a year-round classification, 4 of them are dominated by an anticyclonic anomaly and can be defined as blocked regimes: (i) Atlantic ridge (AR), (ii) European blocking (EuBL), (iii) Scandinavian blocking (ScBL) and (iv) Greenland anticyclone (GL). The methodology is based on the decomposition of the Potential Vorticity (PV) equation into various tendency terms and to project them onto each blocked-regime PV anomaly to identify the terms and processes responsible for the formation of each blocked regime. It follows a similar approach as that done in some previous studies using the relative vorticity equation but is here applied to the PV equation and the decomposition of the terms is different as well. Another originality is that the decomposition of the weather regimes is here based on the recent year-round 7-regimes classification of Grams et al (2017) that provides more refinement than the classical 4-regimes classification of Vautard (1990). The methodology is also based on the decomposition of the blocked regimes onsets onto two types of clusters; one associated with a westward displacement of the anomalies (retrograde cluster) and another associated with an eastward displacement (upstream cluster). Even though it lengthens the paper and the number of figures, the most original part of the results comes out from that last decomposition. It shows that the upstream cluster is mainly dominated by linear Rossby wave dynamics while the retrograde cluster by non linear eddy interactions. Other processes are discussed: baroclinic processes, moist processes, dependence on the season or on the type of regime transition. Overall the paper is well written and well organized, figures are well chosen and clear, and the result on the difference between retrograde and upstream clusters makes the paper particularly interesting to better understand weather regimes formation and transitions. I do not have a major concern about any of the sections of the paper but my list of comments is a bit long and I recommend publication once the authors have adequately addressed them.

We thank Gwendal Riviere for his careful reading of our manuscript and his thoughtful and insightful comments. The comments helped to further improve both, the interpretation and presentation of our results. Our responses

to his comments are given below in blue.

**Major comments**

1. The introduction is well written but in my opinion it does not provide enough information on the various definitions of the blocking. A blocking is indeed a circulation pattern that blocks the mean jet but this might be created by a dipolar anomaly and not necessarily a blocking anticyclone (even though I agree that in the real world the most common situation is the domination of the high). Furthermore, the introduction gives the feeling that all blocked weather regimes, ie. where anticyclonic anomalies dominate following the definition of the paper, have low predictability or are less well represented than the other regimes (see lines 35-38). Is it really true ? Because in my opinion, the Greenland anticyclone, which sometimes refers to negative NAO, is not known to exhibit lower predictability than other regimes. Finally, defining GL as a blocked regime presents that regime as similar to the others. However, it has an effect on the jet which is very different than the ScBL, EuBL or AR. In GL, the jet is shifted to the south but is zonal whereas in the other three regimes, the jets are not straight and there is always an abrupt deviation in the westerlies latitude at some longitudes. I know the paper is quite long but since it is entirely relying on this definition of blocked regimes, some caution should be taken or some warning needs to be provided to the reader to explain that in the present paper blocked regimes refer to the dominance of anticyclonic anomalies.

   Thank you very much for pointing out these aspects of our representation that require clarification. These clarification help to further improve our introduction. There are three aspects in the reviewer's comment: block definition, predictability, and the relation between the blocked weather regimes and jet regimes. We have clarified the first two aspects by splitting the first paragraph of the introduction into two paragraphs. At the end of the (new) first paragraph we now note that "It is important to note that blocking anticyclones do not comprise all possible flow configurations that constitute a block. For example, a dipole anomaly, in which the anticyclone does not dominate, may form a block also. Arguably, however, in the real atmosphere blocking is most commonly associated with a dominant anticyclone." At the end of the (new) second paragraph we now note that "In general, however, the predictability of blocking anticylones is not less than that of cyclone-dominated regimes, and there are differences also in the predictability of different regimes dominated by blocking anticyclones (Ferranti et al., 2018; Büeler et al., 2021). An important question is thus to what extent differences in predictability can be understood in terms of differences in the dynamics that govern the respective regime life cycles."
   The third aspect is addressed in the (new) third paragraph. We agree with the reviewer that it is important to mention the different jet regimes with that the different regimes are associated. To this end, we refer to a previous study by Madonna et al. (2017) and have added (after "classification:"): "Blocked regimes constitute four out of seven regimes in this classification: Atlantic Ridge (AR), European Blocking (EuBL), Scandinavian Blocking (ScBL), and Greenland Blocking (GL). For winter, it has been demonstrated that these weather regimes correspond to different jet regimes: AR with a northern jet, GL with a southern jet, and EuBL with a tilted jet (Madonna et al., 2017)."

2. Section 2.4: I think this section needs to be better inserted within the text. When I read it, I clearly did not understand why there was this discussion on EOF and k-means clustering here. I thought it was a mistake and was there to describe the classification of weather regimes, which in fact was already present in the beginning of section 2.2. So reading section 2.4 was confusing. It is only in section 4.1 that I understood that it is applied to the PV anomalies to separate the weather regimes onsets onto two different clusters: retrograde and upstream. To be understood, section 2.4 needs to start by explaining why we want to do an additional

classification and precise to which variable it is applied. If I understood correctly, it is on PV but this is not said.

Upon reflection, we agree that giving this information at that point in the manuscript may be somewhat confusing to readers. In response to this comment we have moved the information from section 2.4 to the beginning of section 4 to introduce the cluster analysis just before its results are presented.

3. First paragraph of section 3 (lines 267-279). This paragraph is difficult to follow when we do not know what are the two pathways in question. Since the paper is already lengthy and this paragraph did not bring key information for me to understand the next subsections, my suggestion is to significantly reduce it. One or two general sentences to introduce the section would be enough.

Thank you for this good point. We now actually omit this introductory paragraph completely, because we believe that the organization into three subsection makes such an introduction dispensable. In addition, we have changed the title of section three to "Mean perspective on the different blocked regimes".

4. Barolinic term. I struggled a lot to understand the signs of the baroclinic coupling terms. In all the reasonings I have made, I got the opposite signs. The baroclinic coupling term is $-(v'_{low} \cdot \boldsymbol{\nabla} q_0)^L$ while the intrinsic wave propagation term is $-(v'_{up} \cdot \boldsymbol{\nabla} q_0)^L$. Since $q_0$ is a climatological mean, the baroclinic term is roughly $-(v'_{low}{}^L \cdot \boldsymbol{\nabla} q_0)$ and the intrinsic wave propagation can be approximate by $-(v'_{up}{}^L \cdot \boldsymbol{\nabla} q_0)$. Since the low-frequency anomalies are mainly barotropic equivalent we expect $v'_{low}{}^L$ and $v'_{up}{}^L$ to have roughly the same signs (even though there might a slight westward tilt with height). Hence, I would expect the baroclinic coupling term to be negative upstream and positive downstream of the anticyclonic anomalies but Figures 6a-d clearly show the opposite. Could you check the signs or show the circulation associated with $v'_{low}$, or show a section of the circulation across the negative PV anomalies? I am pretty sure that the patterns of the baroclinic coupling term can be easily interpreted and should be presented in the paper. One possibility would be to draw a schematic to show in the paper.

We have double checked the baroclinic tendencies and their sign is correct. As the reviewer notes in his comment 14, it is consistent with an approximate equivalent barotropic structure i) to find the temperature anomaly approximately below the upper-level anomaly and ii) for the temperature anomaly to have the same sign (a warm anomaly) over the whole column. The counter-intuitive aspect here seems to be that the low-level anomaly constitutes a positive PV anomaly, whereas the PV anomaly at upper levels is negative. It is important to note, however, that the (weak) low-level positive PV anomaly does not need to imply a cyclone structure at low levels (cyclonic winds and a negative anomaly in geopotential). Low levels may still be characterized by high pressure and anticyclonic flow because the relatively strong upper-level PV anomaly may (and in fact does) dominate over the relatively weak low-level anomaly. It is not entirely clear to us to what extent discussion of this potential source of confusion is helpful rather than confusing to most readers. We thus opt for adding a brief footnote that addresses the issue: "Note that despite the low-level warm anomaly, which implies a positive PV anomaly in the northern hemisphere, the (total) low-level structure in wind and geopotential may still exhibit anticyclonic characteristics, because the strong negative upper-level PV anomaly may dominate over the relatively weak low-level anomaly."

5. The discussion is interesting but is quite long (2 pages) and centred on what the same team has published in the recent past. There is no really a discussion on what are the new aspects on blocked regimes the paper brings out and what are the results confirming previous studies.

In response to another reviewer, we have extended the discussion on seasonality, including a season-mean (extended summer and winter) perspective on variability. With this discussion, it now should become clearer that the main result of this study is the identification of a mode of variability that is arguably more fundamental

[Figure]

Figure 1: **Composite maps of low-frequency PV anomalies (shading) for GL after onset.** a) Blue contours depict zonal wind anomalies at same isentropic levels as PVa (for $\pm$ [3,6,9,12,15] m/s). b) Magenta contours depict the anomalies of synoptic-scale Rossby wave envelope relative to climatology (for $\pm$[1,2,3] m/s). Grey contours depict the PV regime pattern (for $\pm$[0.2,0.4,0.6,0.8] PVU, negative values dashed).

and physically more meaningful than a description of variability that is based on seasonal means. In addition, in response to all reviewer comments, we have included four more references to previous studies, which are without contribution by one of the co-author, into the concluding discussion. We thus hope that these revisions now provide an acceptably balanced discussion.

(a) There is only one sentence (in the summary 5.1) where the authors mention that GL is consistent with the traffic jam theory of Nakamura and co-authors while the other three regimes are not. I am a bit surprised that this is precisely GL that fits the traffic jam theory because it corresponds to a high-latitude blocking that cannot really stop the mid-latitude wave propagation. Because the other three regimes present a mid-latitude blocking, they would have been better candidates for the traffic jam theory.

Minor comment 10 is related, we address both here. GL indeed shifts the jet in terms of zonal wind anomalies to the south (Figure 1a). However, the anomalies associated with the shift are not of equal strength, resulting in an overall weakening of the zonal wind over the North Atlantic. Together with the clear signal of negative envelope anomalies downstream of the GL anticyclone (Figure 1b), we obtain a picture of reduced wave propagation consistent with the findings from the LWA framework. We are therefore confident about the robustness of our findings. The subsequent association of the GL LWA flux anomalies with the traffic jam model of Nakamura and Huang (2018) in our work is primarily of a descriptive nature. The absence of a positive LWA flux anomaly north or south of the suppression region, as seen for the other regimes, makes GL a particularly good fit for the one-dimensional traffic jam description of blocking.

A few features of the LWA/traffic jam framework may relate to the concerns raised by the reviewer and are mentioned here. Wave propagation along the mid-latitude waveguide is considered with respect to the background state waveguide, which in the framework exists in the "zonalized" background state.

This zonally-symmetric state does not allow for longitudinal variation and a local feature over the North Atlantic may not cause a significant meridional shift of the waveguide. Further, PV anomalies quantified in terms of LWA are attributed to the equivalent latitudes of the PV contours evaluated in the computation, such that PV anomalies can translate to LWA at lower (or higher) latitudes. In a GL composite we find the LWA anomaly associated with the regime at similar latitudes than the climatological LWA maximum found over the European continent.

We do not wish to expand significantly on the traffic jam interpretation in the manuscript but have made changes to the phrasing in both the summary and section 3.1. In section 3.1, we now specify that the zonal propagation of wave activity is suppressed "along the background state jet, acting as a 'waveguide'", to clarify that in the traffic jam framework the background state zonal wind $U$ is "waveguide-defining". In the summary, we now state that GL is "*most* consistent with the 'traffic jam' *description* of blocking". We have also changed the wording in the following sentence on the deflection-type regimes, now stating that this aspect is "not reconcilable in an obvious way" with the traffic jam theory, when before we claimed that it "cannot be represented". These changes highlight that our interpretation is based on descriptive features, rather than a detailed evaluation of the Nakamura and Huang theory. Blocking events attributed to non-GL lifecycles may also be describable by the traffic jam framework. However, based on our results, such descriptions are likely complicated by the meridional dipolar structure in the LWA flux anomalies which the one-dimensional theory does not accommodate in a straightforward way, but which is absent during GL.

(b) Also since the paper highlights the linear vs nonlinear processes in driving regimes I would expect more discussions with regard to the results of Michel and Riviere (2011) who have also focused on the relative importance of linear vs nonlinear processes in the formation of weather regimes. For instance, Michel and Riviere (2011) found that linear processes first trigger the formation of WR and then nonlinear processes associated with wave breaking play an important role in reinforcing the WR. In the present paper, I would have expected the quasi-barotropic linear dynamics to be first dominant and then followed by the non linear eddy term. But figure 7 does not show it (I do not see any time lag between the two terms) and this is puzzling for me. Is it because the quasi-barotropic linear term does not contain all the linear terms (it should be added to the baroclinic one)? Is it because the non linear eddy term does not include all the non linear terms (it misses the term with divergence $-v'_{div} \cdot \boldsymbol{\nabla} q'$)?

Good point. In fact, we have compared our results to Michel and Rivière (2011, MR11) before submission and a preliminary version of the manuscript had included a discussion of this comparison. This discussion, however, turned out to be lengthy and unfortunately little fruitful, because - as noted by the reviewer, the results of our study and MR11 are rather different, but so are the methods applied. Arguably, the differences in the applied methods render a direct comparison unfeasible. The main differences in the methods between MR11 and our study are: i) The definition of onset time is different between MR11 and our approach. ii) MR11 used data from extended winter (with winter regime definitions) while we consider year-round regime dynamics. iii) MR11 applied a streamfunction tendency equation with a partitioning of tendencies into high- and low-frequency components, we use PV tendencies and only focus on low-frequency components. iv) MR11 stratified their periods according to transitions between regimes, whereas we include all transitions. v) The separation between linear and nonlinear dynamics is different, as noted by the reviewer: The linear dynamics in MR11 would contain not only our QBBG-term, but also BCBG and linear components of DIV $(-v'_{div} \cdot \boldsymbol{\nabla} q_0)$, while the nonlinear term would not only consist of EDDY but also of components of DIV $(-v'_{div} \cdot \boldsymbol{\nabla} q')$.

We had found the greatest potential for direct comparison between the transition from AR to GL in MR11 (their Fig.8e) and our GL retrograde cluster (Fig.9d), because this cluster contains a large portion of transitions from AR (not shown). For this case, our results are consistent with those of MR11, in the

sense that both studies show the dominance of nonlinear dynamics and subordinate linear dynamics. As indicated above, referring to this agreement between results would necessitate a rather lengthy discussion of differences in methods, and the resulting agreement is then only a rather specific one. We are not sure, if the manuscript would benefit from such a discussion. We should not have, however, completely omitted a reference to MR11's results. In the summary, we now add such a reference, which reads: "Using a piecewise-tendency framework for the streamfunction, a framework that is in general similar to our approach, Michel and Rivière (2011) found that linear dynamics lead the formation of a weather regime and subsequently nonlinear processes reinforce the regime. We do not see this signal in our study. Unfortunately, however, a number of more specific differences between our framework and that of Michel and Rivière (2011) prohibit a direct comparison of results[1]. It would be very interesting to reconcile both results, but a separate study focusing on this reconciliation would be needed to do so."

**Minor comments:**

1. Line 144: The text says that a 90-day running mean is applied but in appendix a 30-day running mean is applied. Why is there a difference? The calculation of the geopotential anomalies for the weather regimes and the PV anomalies for PV inversion and PV-tendencies use slightly different background fields. Both are based on a daily climatology, i.e., daily averaged over the ERA5 period under consideration, but for the geopotential anomalies the daily climatology is smoothed by a 90-day running mean to mimic a seasonal climatology and for the PV anomalies by a 30-day running mean to mimic monthly means. The differences are hence only in the smoothing. We used a weaker smoother since the PV tendencies, which are quite expensive to calculate due to nonlinear PV inversion, are not only used for low-frequency considerations but for other studies also.

2. Line 145: a reference for k-means clustering is needed and maybe a sentence describing it. Reference added in L391 within Sec.4

3. Line 160-169: the number of cases selected by the described procedure should be provided. How many cases or days correspond to the composites of Figure 1? Thank you for pointing out this miss. We added the number of cases within the figure as reference.

4. Line 185: please indicate here that the derivation of that equation is provided in Appendix A. reference to appendix can be found in L218

5. Line 229: Why is the computation of envelopes of synoptic Rossby waves important here? Since Fig.3 shows a simple composite, I would expect a simpler diagnostic of synoptic Rossby wave to be also relevant like EKE (eddy kinetic energy where eddy contains the synoptic part only). Is there a reason for using such a sophisticated diagnostic here?
We agree with the reviewer that other diagnostics could be used here and we, too, expect that EKE composites would show consistent results. A detailed discussion of different diagnostics for RWPs can be found in Wirth et al. (2018). In our group, however, there is extensive experience with RWP envelope analyses and this diagnostic is easily available to us. It is thus for pragmatic reasons that we prefer this diagnostic over other options. We have added a brief comment in this regard to the end of Sect. 2.3.3.: "Several frameworks are available to sensibly diagnose Rossby waves packets (reviewed, e.g., in Wirth et al. (2018)) and we would expect other diagnostics to yield consistent results. The envelope metric is easily available to us and is thus employed in this study."
* * *
[1]These differences are detailed in our responses to the reviewers (cite this response)

[Figure]

Figure 2: **The revised version of Figure 3 of the manuscript, with new contour intervals.** Composite maps of low-frequency PV anomalies (shading). Upper row (a-d): before onset (averaged from 3 days to 1 day before onset). Bottom row (e-h): after onset (averaged from $1 - 3$ days after onset). The respective regime is given at the top of each column. Magenta contours depict the envelope of synoptic-scale Rossby waves (for $\pm[15,18,21]$ m/s). Grey contours depict the PV regime pattern (for $\pm[0.2,0.4,0.6,0.8]$ PVU, negative values dashed).

6. Line 235-241: as mentioned above, I would expect more details on the objectives of that subsection and also the variable(s) to which is applied the algorithm Following your previous comment we decided to move this paragraph into Sec.4., where more background information is given.

7. Line 267-279: as said above, this paragraph is really too long and too general to be understood by the reader. We agree. Please see our response above.

8. Figure 3: Could you explain the choice of the values for the magenta contours ? 16, 18 and 20 m/s are very close to each other and values a bit below 16 m/s could be relevant as well. For instance, why not 10, 15 and 20 m/s ? First, we need to note that absolute values of our envelope metric have changed (by approx. 10%) between the manuscript and the revised version due to a minor bug that we have found in the wavelength filter of the envelope calculation. Our previous contour intervals of [16,18,20 m/s] now translate approx. to [18,20,22 m/s]. In general, the contours have been chosen according to values of the envelope climatology. Within this climatology values below approx. 15 m/s (new values) can essentially be considered as a background signal and are thus little representative for Rossby wave packets. Our choice of contours had thus focused on the core regions of wave packets. In response to your next comment below, we have changed our contour interval to [15,18,21 m/s]. Including the 15 m/s contour indeed reveals an important signal that had previously been concealed. Thank you for the comment! The revised version of Figure 3 of the manuscript is shown below as Figure 2. According to the revised version of the figure, we have somewhat revised (and shortened) the accompanying discussion, which now reads: "With the Rossby wave envelope, we find that large values of the envelope extend into and over the anticyclonic regime anomaly for AR and EuBL after onset (Figure 2 e,f), whereas this signal is much less clear for ScBL and GL (Figure 2 g,h). A comparison

with the envelope before onset (Figure 2 a-d) reveals that the envelope extends downstream during onset for AR and EuBL whereas it retracts (upstream) during regime onset for GL, with a less clear signal for ScBL. Madonna et al. (2017) demonstrated a connection between the meridional jet locations and certain regimes: AR is associated with a northern jet location, EuBL with a tilted jet (southwest to northeast), and GL with a southern jet location. For ScBL, some continuity between the western North Atlantic storm track and the northeastern branch of the jet has been documented in Michel et al. (2012). The signal in our envelope metric, which indicates "waviness" along a jet, is consistent with these jet characteristics after regime onset (Figure 2 e-h)."

9. Line 292-294: I am a bit surprised by the fact the northern branch could not be a continuous extension of the North Atlantic waveguide. For the transition from zonal to ScBL regime studied in Michel et al (2012, GRL), some continuity is shown between the western Atlantic storm-track and the northeastern edge. I see two explanations. One is the fact that the present study considers all transitions to ScBL and not only one. The other is that the northern branch can be seen by lowering the magenta contour interval in Fig.3. Thank you very much for pointing out this apparent inconsistency. Indeed, the northern branch of ScBL can be identified when including smaller values of the contours as shown in the new version of Fig3 (Figure 2). We have added a reference to Michel et al. (2012) and changed the text accordingly (see response to previous comment).

10. Line 297-298: I am surprised that GL comes out as the illustration of the traffic jam theory because, as said before, the whole jet is shifted southward and there is no interruption of the westerly wave guide in such a regime. See our response to major comment 5a.

11. Figure 5: dashed blue contours are not visible. contour style changed for negative values

12. Line 320-326: the discussion on the existence or not of wave trains as function of the WR sounds a bit unfruitful for me. EuBL, AR and ScBL have more zonally oriented anomalies while GL is structured with a meridionally oriented dipolar anomaly. So the PV tendencies reflect more the structures of WR they are building rather than the existence of wave trains in my opinion. We agree with the reviewer that the pattern of the PV tendencies reflects the structure of the weather regime. We meant to say that the structure of EuBL, AR, and ScBL are consistent with a wave packet, whereas GL is not. To us, this observation seems notable in the context of the relation between wave packets and blocking suggested in Yeh (1949), and in particular the renewed interest in this idea by Lei Wang (unfortunately in unpublished form only). To clarify our interpretation and to provide better context, we have modified the text as follows (changes in bold): "Importantly, both patterns resemble a wave **packet that extends beyond the dominant anticyclonic anomaly of the regime pattern** for EuBL, AR, and ScBL (exemplified for EuBL in Figure 5a), whereas for GL (Figure 5d) the anticyclonic anomaly dominates the regime pattern flanked by a zonally oriented cyclonic anomaly to the south. Our interpretation of these differences is that **the structure of** EuBL, AR, and ScBL **can be considered to be consistent with that of** a larger-scale, low-frequency wave packet, whereas **the structure of** GL is not**, largely consistent with our analysis of wave activity flux anomalies above. A relation between Rossby wave packets and blocking has first been suggested by Yeh (1949) and has recently been found renewed interest (Wang and Kuang, 2019).**"

13. Line 335-338: The effect of the dipolar anomaly on the reduction of the mean PV gradient is difficult to appreciate because the mean PV gradient is not shown and the dipolar anomaly is tilted (see Fig5c). Is it really useful to say that here? It is true that an estimated axis of the dipole is not perfectly perpendicular to the background PV (Figure 3). However, it seems evident to us that parts of the dipole structure project strongly on the PV background gradient upstream of the anticyclonic anomaly, as indicated by the black lines

[Figure]

Figure 3: **Fig05c,f with PV background for all blocked regimes.** Composite maps of low-frequency EDDY tendencies (shading) after onset for a) AR, b) EuBL, c) ScBL and d) GL. Black lines indicate dipole parallel to PV gradient included by hand for illustrative purpose. Green contours depict the PV background (for [2,3,4,5] PVU). Grey contours depict the PV regime pattern (for ±[0.2,0.4,0.6,0.8] PVU, negative values dashed).

in Figure 3. We believe that there is a robust signal here, which is in line with previous literature and it seems worth mentioning to illustrate how the EDDY-term influences blocking formation. However, in favour of visual clarity we would like to omit PV background contours in the manuscript. Instead, we have added "not shown" in the manuscript in L336.

14. Line 336-346 and Figure 6: the fact that a low-level positive temperature anomaly lies below the anticyclonic anomaly is consistent with the fact that the anomalies are barotropic equivalent and dominated by upper-levels. Indeed, an upper-level anticyclone with amplitude decreasing as it goes closer to the surface is associated with a warm anomaly over the whole column because the sign of the vertical gradient of the geopotential is the same as the sign of temperature following hydrostatic relation. As mentioned above, this gave me the wrong sign for the baroclinic coupling term. So I would need clarification on that aspect. Please see our response to your major comment 4.

15. Line 385-386: usually in k-means classification, the choice of the number of clusters will dependent on the ratio between inter and intra variances. Does the number 2 appears as the most appropriate number? As mentioned in section 2.4. L238-241 we checked common heuristics to define an optimal number of clusters. However, no optimal cluster could be found regarding these classifications. Furthermore we looked into clusters up to seven clusters searching for variability associated with e.g. season (four clusters) or weather regime transitions (seven clusters) and found no sufficiently sharp signals. Since we are interested in the main variability of the dynamics, we hence decided to use two clusters. We have moved information from subsection 2.4 to the beginning of section 4. This somewhat revised information now reads: "Using common heuristics[2] to determine the optimal number of clusters did not yield unambiguous results. We have performed preliminary analyses with four and seven clusters to explore if variability would be predominantly associated with individual seasons or regimes transitions, respectively. With these cluster numbers, however, several of the cluster-mean patterns did not appear to be sufficiently distinct for further in-depth analysis. Because our interest here is on the leading-order variability of regime dynamics, we simply use two clusters for the k-means clustering."
* * *
[2]We have inspected the described variance as a function of cluster number ("elbow plot"; Thorndike, 1953) and the ratio of intra- and inter-cluster variance (Caliński and Harabasz, 1974).

16. Line 393: Is the sum of retrograde and upstream clusters equal to the number of cases shown in Fig.1 ? Yes. We have added the information in L403: "The sum of both clusters is equal to the total number of cases for all regimes."

17. Line 425-428: Sentence is difficult to fully understand. Thank you for catching this long and convoluted sentence. We have rephrased as follows: "The dominance of this term indicates the importance of advecting pre-existing anomalies into the core region of the regime pattern. The generation and amplification of these anomalies predominantly occur in regions where the regime pattern has small amplitude, and are thus poorly captured by tendencies projected onto the regime pattern. In a case study, Hauser et al. (2022a) used a semi-Lagrangian (anomaly-following) framework to describe this situation explicitly for amplification due to upper-tropospheric divergent outflow."

18. Line 430: the structure of the sentence in the parenthesis is a bit strange. It is. We have rephrased as follows: (Compare the spatial distribution of PV anomalies (Figure 8) with that of the divergent tendencies (Figure 12 and discussion in subsection 4.3).)

19. Line 491: the fact that baroclinic coupling contributes negatively is strange following the reasonings I have made (see above). I am probably wrong...but it needs clarification. Please refer to our response to your major comment 4.

20. Line 493: What is meant when the authors say "located outside the regime pattern" ? Does it mean that the amplitude of the divergent tendencies are far away from those of the regime ? or does it mean divergent tendencies are in quadrature ? Thank you, good catch. Figure 12 does not support a standard interpretation of 'outside' of the pattern. The modified version now reads (changes in bold): "We note that all blocked regimes exhibit a distinct minimum of divergent tendencies adjacent to the negative PV anomaly of the regime pattern. **The tendencies, however, are mostly located in regions where the regime pattern has small amplitude.** The importance of these divergent tendencies **may thus be** underrepresented by the projected tendencies." We have also adjusted the wording in the abstract.

21. Line 536: same question as in 20) We have rephrased similarly as in 20) (changes in bold):... has maximum amplitude **where the regime pattern has small amplitude** and is thus poorly represented by ...

22. Line 554-555: the fact that GL (or negative NAO) is formed via local processes has been emphasized by other studies (see Feldstein, 2003; Benedict et al. 2004) Thank you for drawing our attention to this point in both studies. We included a reference in the discussion: "The rather local evolution of GL (in terms of negative NAO) was also found by Feldstein (2003) and Benedict et al. (2004)."

**2    Response to Reviewer 2**

This paper builds on previous work by the authors and others to examine how different categories of blocking are related to different kinds of forcing, derived from PV tendency analyses. It presents interesting new insight on the apparent similarity of the different categories of blocking, is clearly written and has good diagrams. My recommendation is to publish with minor corrections.

We thank the reviewer for his insightful comments that helped to further improve our manuscript. Our responses to the comments are given below in blue.

**Points for the authors to consider:**

1. l.76 'A distinct advantage of projections is, however, that variability in terms of mere geographical location of different patterns is eliminated and thus the variability of dynamical processes is more directly intercomparable.' How does this statement (repeated on l. 470) tally with equation 2, where the weather regime index is calculated by correlation of an instantaneous pattern to the weather regime patterns, the latter geographically fixed in terms of $\lambda$ and $\varphi$?

   Our statement meant to refer to the fact that the projection focus on the **relative location** of anomalies and PV tendencies to the regime pattern. That means the mere geographical location of the anomaly or tendency is less important than the location relative to the regime pattern. Equation 2 shows this fact by the pattern correlation. Because the projection focuses on the location relative to the regime pattern, the evolution of different regimes (e.g., GL vs. EuBL) are arguably more directly comparable when using projection than when taking into consideration the (absolute) geographical distribution of anomalies and tendencies. We have revised the statement as follows (changes begin after the first "that": "A distinct advantage of projections is, however, that they focus on the location of processes *relative* to the regime pattern, and thus make more directly intercomparable the processes associated with the different weather regimes that occur in different geographic regions."

2. Fig 2. The most striking thing to me about fig 2 is that the four coloured traces are almost identical, yet the authors do not mention this in lines 259-265 where the figure is discussed. No doubt I have missed something very obvious in the previous discussion, but I can't be alone in this, so please could the discussion be extended to explain why the coloured curves are so similar?

   We agree with the reviewer that it is interesting to note that all four regimes evolve very similarly in terms of their projected PV anomalies (as seen by the coloured lines). Following Hochman et al. (2021, their Fig2, esoecially their Fig2h) this behaviour was not unexpected and in fact, we noted the similarity in a short sentence in L261-262. In Hochman et al. (2021) the similarity in the mean projection stems from the definition of the weather regime index (IWR) and the underlying weather regime life cycle definition. First, the IWR is standardised in order to make it intercomparable amongst the regimes. Thus regimes on average share a similar range of values of the projection. Second, the regime life cycle definition filters out periods which last less than 5 days. On average a regime life cycle lasts 9-11 days with some variability between regimes. The surprising result here is that the projection PV anomalies indicates a similar ramping up into the pattern and decay after about 10 days for all blocked regimes. The purpose of Fig.2, however, is to provide validation of our diagnostic, so we refrain from an in-depth discussion. It seems worth to emphasise the similarity a bit more strongly (by putting this observation upfront and citing Hochman et al. (2021) and we have therefore modified the text as follows: "As shown by Hochman et al. (2021, their Fig2) for the original Z500-based standardised projection, the evolution of all four regimes (in terms of the projected PV anomalies, colored lines in Fig02 is largely similar, which stems from the life cycle definition and the average duration of about 10 days for all different blocked regimes. The associated observed tendencies are, ..."

3. L.355-361. This paragraph does not explain how projecting PV tendencies on to the regime pattern minimises the impact of geographical differences, and I can't see any obvious reason why it should. Explain!

   We have added, after the references "(Feldstein, 2003; Michel and Riviere, 2011)": ", because the projections focus on the location of processes *relative* to the regime pattern." Please refer also to our answer to your first comment.

**Typos and minor corrections**

- l.59, 365 remove comma done

- l.85 exceeds done

- l.102 regimes done

- Fig 1 caption: Year-round done

- l.184 Brackets incorrectly applied around reference. Also similar to (not as) done

- l.203 nonconservative done

- l.215 define LWA (I presume it's the same as A, the local finite-amplitude wave activity) done

- l.258 Brackets incorrectly applied around reference.done

- l.292 locations done

- Fig 5 caption: explain that top row is EuBL and bottom row GL. The left hand panels confused me because (I think that) negative blue contours are shown as solid, same as the positive ones (i.e. all contours are positive). According to my reading of the text l.327, some of them should be negative. Yes, unfortunately negative values have not been dashed. We changed that in the new figure.

- l. 380 'either' suggests that there are only two pathways, yet that is not how the text reads We rephrased the sentence accordingly: "If distinct pathways to onset existed, however, the average picture would not be representative of any of these pathways."

- l.456 regime's done

**3 Response to Reviewer 3**

This paper provides interesting insight into the dynamics of several regimes of Euro-Atlantic blocking. The paper demonstrates how the various regimes have certain similarities and differences with regards to the evolution of potential vorticity and its tendency forcing terms as well as local wave activity. This paper is useful in helping to reaffirm blocking dynamics can be sensitive to geographic location and transient and mean state flow features. I recommend minor corrections
We thank the reviewer for the insightful comments, which inter alia helped to clarify our perspective on the relation between seasonality and the main modes of variability described in this manuscript, and thus were very helpful to further improve the manuscript. Our responses to the reviewer's comments are given below in blue.

**Points for the author to consider:**

1. L. 113-114 I was actually surprised to read that seasonality did not play much of a role. I have always worked under the assumption that summer vs. winter blocking, for example, behaved differently. For example, Winter blocking could be more influenced by stronger stationary waves and eddy activity compared to Summer blocking which exists in a moister background and weaker zonal flow.

   Thank you for this comment. Other readers may wonder about the role of seasonality early on in the manuscript also. It may thus be helpful for readers to hint to the more detailed discussion of this issue later in the manuscript, in section 4.2 and in the discussion section, already in this introductory paragraph. (Section 4.2 has been revised in response to your next comment.) In response to your comment we have revised the text in L113-115, including some minor changes in wording. The revised version now reads (changes in bold): "A posteriori, the approach is justified because **it yields the significant result** that the main modes of variability do not **merely reflect** differences in season (e.g., extended summer vs. extended winter) or

[Figure]

Figure 4: Number of regime life cycles for a) MAM, b) JJA, c) SON, and d) DJF for both clusters and the 4 blocked weather regimes.

preferred regime transitions. **The relation to seasonality and regime transitions will be discussed in some detail in section 4.2 and the discussion section, respectively.**"

2. You claim that the main modes of variability do not depend on season and this is discussed in the discussion section, but I have some questions about the downplay of seasonality. Is one still able to capture the same blocking cluster patterns/regimes if the regime finding algorithm is applied to winter only vs. summer only? Also, does the evolution of the PV anomaly, envelope and LWA before and after onset matter with regards to season? Perhaps the answer to the latter question can be addressed in a supporting information section.

It is important to us to clarify that we did not mean to downplay seasonality. In fact, the discussion section had been very explicit about the seasonal variation in the occurrence of the clusters. We understand, however, your concern that this discussion may have been too brief or not sufficiently motivated. We hence give a detailed answer here addressing the following points: i) downplay of seasonality, ii) regime behaviour between seasons, and iii) PV, Envelope and LWA for winter/summer.

i) Downplay of seasonality:
Figure 4 shows the distribution of cluster occurrence with season. Without showing the figure, it has been discussed in the discussion section. The seasonality of ScBL stands out, which we have highlighted in the manuscript already. Furthermore, we have already noted the predominance of the retrograde cluster for EuBL in winter, which shows rather prominently also. To give the discussion of seasonality more weight in the manuscript, we will add a reference to this figure in this response in the revised discussion.
The important point that we make in the discussion is that the two different clusters do not merely reflect differences between seasons (or types of regime transitions). Your comment demonstrates that it will be helpful to readers to further extend discussion on seasonal variability. To do so, we add to the revised version the season-mean dynamics of the blocked regimes, as diagnosed by the projected piecewise PV tendencies, in direct comparison to the cluster-mean dynamics (in the revised version of Fig. 9, here presented as Figure 5). The mean dynamics of extended winter vs. extended summer (cf. Figure 5 i-l to m-p) show differences regarding the linear and nonlinear quasi-barotropic dynamics (QBBG and EDDY) and regarding baroclinic coupling (BC). Overall, however, it is clear that the differences between the extended seasons are less pronounced than those between the two clusters. The revised version of Section 4.2 now includes a discussion of Figure 5. The first paragraph of this subsection now reads: "The evolution of the dynamical mechanisms governing the formation of the regime pattern for the two clusters for all four regimes are shown in the first two rows in Fig. 9. We additionally show the evolution for extended winter (November-March; NDJFM) and extended summer (May-September; MJJAS) (last two rows in Fig. 9). Evidently, for all regimes, the two different clusters exhibit

[Figure]

Figure 5: **New Fig09 with seasonal differences included.** Piecewise PV tendencies projected onto the respective regime pattern for $\pm 5$ days around onset: left column - AR, second column - EuBL, third column - ScBL, and right column - GL. Dynamics shown for both clusters (top row - retrograde, second row - upstream) and both extended seasons (third row - NDJFM, bottom row - MJJAS). Colors represent different processes: Blue: Quasi-barotropic dynamics, dotted if intrinsic propagation (WAVE) dominates advection by the background flow (ADV), solid otherwise. Yellow: Baroclinic coupling. Red: Divergent tendency. Green: Eddy flux convergence.

very different dynamics. Importantly, these differences are much more pronounced than i) the differences between the individual regimes in the mean sense (Fig. 7), ii) the differences between the individual regimes within the retrograde and the upstream clusters, respectively and iii) the differences between the extended seasons." The more detailed discussion of the figure reads: "Between the extended summer and winter seasons (Fig. 9 i-p), one difference for all regimes is the larger contribution by baroclinic coupling in winter, which can be expected due to generally stronger baroclinicity in winter, and which has been found also in the context of Rossby wave packet dynamics (Teubler and Riemer, 2021, their Figure 7). Besides differences in baroclinic coupling, summer and winter dynamics – in terms of the projected piecewise PV tendencies – are most similar for GL. For AR, linear dynamics contribute more strongly than nonlinear dynamics during summer, whereas the contributions are similar during winter. In contrast, for ScBL the linear dynamics contribute more strongly than nonlinear dynamics during winter, whereas the contributions are similar during summer. For EuBL, the nonlinear dynamics appear to lead the linear dynamics, which appears more prominently during winter when the linear dynamics are somewhat stronger. Overall, however, as noted above, the differences between the extended summer and winter seasons are evidently less pronounced than between the retrograde and the upstream clusters, at least in the framework of the projected piecewise PV tendencies. Notably, besides the role of baroclinic coupling, we do not find systematic differences between seasons that would apply for all four regimes. This result suggests that for the year-round weather regimes the dynamics-based variability described by the retrograde and upstream clusters represent a more fundamental mode of variability than a description of variability that is based on the comparison of seasonal means."

ii) Regarding the weather regimes in different seasons: You are correct that the regimes will be different if a cluster algorithm is applied to individual seasons only. The year-round definition is one valid and valuable approach to define regimes, though. Differences between the regimes regarding season are noted in the introduction (L102-117). We investigate the main variability in the dynamics of these year-round regimes. We believe that it is clear from our introduction that this is the goal of the current study.

iii) We here present the distribution of PV anomalies, the Rossby wave envelope, and local wave activity fluxes for summer and winter separately (Figure 6, Figure 7, Figure 8, and Figure 9). The pattern of PV anomalies after onset is very similar between summer and winter. Some differences occur before onset, again most pronounced for ScBL. AR and GL are most similar for both seasons. Differences can be expected due to the seasonal variation of cluster occurrence documented in the discussion section. Overall, however, the differences between seasons appear to be smaller than those between the clusters (cf. Fig08 of the manuscript).
The envelope is much weaker in summer than in winter. Similar to the year-round mean (Fig03 of the manuscript) for both winter and summer the envelope mainly retracts for ScBL and GL after onset, while the envelope for EuBL and AR is rather deflected. Similar to the envelope, the LWA flux patterns associated with the regimes are substantially weaker in summer than in winter. However, the general pattern after onset is preserved, although there are some differences like the stronger asymmetry in terms of amplitude of the dipole for AR and ScBL.
Differences between summer and winter do occur, and we never meant to downplay these differences. The revised Fig. 9 and the numbers (and reference to Figure 4) given in the discussion section clearly demonstrates these differences. The point we meant to make in the initial version of our manuscript was that the variability that we found in the retrograde and upstream cluster – starting from the characteristics of dynamical processes alone – dominates and transcends the (presupposed) variability with season (and types of transitions, not shown). Extending Fig. 9 in the revised version helps to make this point more clearly. We refrain, however, from including a more detailed analysis of differences between seasons in the current study.

3. L. 490-491 How are the results of baroclinic coupling not playing a constructive role consistent/compatible with the findings of Martineau et al. (2022, GRL, Baroclinic Blocking)? They found that winter blocking

[Figure]

Figure 6: **Fig03 for DJF.** Composite maps of low-frequency PV anomalies (shading) for DJF. Upper row (a-d): before onset (averaged from 3 days to 1 day before onset). Bottom row (e-h): after onset (averaged from $1-3$ days after onset). The respective regime is given at the top of each column. Magenta contours depict the envelope of synoptic-scale Rossby waves (for $\pm[16,19,22]$ m/s). Grey contours depict the PV regime pattern (for $\pm[0.2,0.4,0.6,0.8]$ PVU, negative values dashed).

[Figure]

Figure 7: **Fig03 for JJA.** Same as Figure 6 but for JJA with smaller contour intervals ($\pm[14,17,20]$ m/s).

[Figure]

Figure 8: **Fig04 for DJF.** Same as Figure 6 but for low-frequency anomalies of the zonal local finite-amplitude wave activity flux $F_\lambda$ (shading).

[Figure]

Figure 9: **Fig04 for JJA.** Same as Figure 8 but for JJA. Note different colorbar.

over Greenland had strong contributions from baroclinic energy extraction from the mean state.

Thank you very much for pointing to this relevant study. In response to this comment, we have extended the discussion in L492, which now reads: "We note that all blocked regimes exhibit a clear pattern of baroclinic tendencies as well as a distinct minimum of divergent tendencies adjacent to the negative PV anomaly of the regime pattern. Both baroclinic and divergent tendencies, however, are mostly located in regions where the regime pattern has small amplitude. The importance of these tendencies, which in combination signify moist-baroclinic growth, may thus be underrepresented by the projected tendencies. Martineau et al. (2022), considering a larger domain than the regime pattern, find that baroclinic energy conversion is a major energy source for blocking over Greenland during winter. We believe that the small extent to which the baroclinic and divergent tendencies project onto the regime patterns explain the qualitative differences between our result and that of Martineau et al. (2022)."

In addition, we have added a footnote that provides some more information why a direct comparison between their and our results may not be straightforward. The footnote reads: "It is important to note that the energy and the PV frameworks are not directly comparable (discussed in some detail in Teubler and Riemer, 2016; Wirth et al., 2018). For example, diabatically enhanced ascent in warm regions usually contributes positively to baroclinic conversion, whereas this ascent is predominantly captured by the divergent term in the PV framework. In addition, the signal in Martineau et al. (2022) maximizes in the lower to mid-troposphere whereas we analyze PV anomalies in the tropopause region. We believe, however, that the qualitative difference between Martineau et al.'s and our results is due to the location of the tendencies relative to the regime pattern, as noted in the main text."

In addition, we now refer to baroclinic growth also when we discuss in some more detail the variability of moist processes in Sect. 5.2.2. In L537 we add after "moist processes": "and of moist baroclinc growth". In the abstract also we now refer to "moist-baroclinc growth".

Finally, we have added a reference to Martineau et al. (2022) in the introduction when we first refer to the role of baroclinic growth for blocking.

4. Figure 7. It is unclear to me why the units for Piecewise PV tendencies are day-1 and not the same as the PV tendences you show in the previous figure.

In Figure 7 the piecewise PV tendecies are quantified by projecting the tendencies onto the regime pattern following Equation 8 (here repeated)

$$P_{\partial q^L/\partial t}(t)\big|_p := \frac{\sum_{(\lambda,\varphi)\in NH} \frac{\partial q^L(t,\lambda,\varphi)}{\partial t}\big|_j \, q^L_{WR}(\lambda,\varphi)\,\cos\varphi}{\sum_{(\lambda,\varphi)\in NH}(q^L_{WR})^2(\lambda,\varphi)\,\cos\varphi}, \qquad p \in \text{WAVE, ADV, BC, DIV, EDDY}.$$

The units are hence $[PVU/day \cdot PVU/(PVU^2)] = 1/day$.

5. Also, is there a budget that these terms should add up to? If so, would the residual of that budget be linked to moist processes and friction? A budget for LWA is used by Neal et al. (2022) where the residual seems to be directly linked to moist diabatic contributions to blocking.

The PV tendency equation (equation 4) defines a budget, whose residual (the difference between the diagnosed (rhs) and observed PV tendencies (lhs)) is shown in Fig. 2 in Section 2.5. As mentioned in the Appendix A the residual also contains the missing nonconservative tendencies $\mathcal{N}'$ (the deviation from the climatological mean). In the case study of Hauser et al. (2022b), we estimated nonconservative PV tendencies from temperature and wind tendencies from ERA5 short range forecasts and showed that the residual cannot be closed completely by the use of nonconservative tendencies. We hence would not like to use the residual as estimator for nonconservative processes. Additionally our divergent PV tendencies are closely associated with latent heat

release below as also was explicitly shown in Hauser et al. (2022b) and serve in our diagnostic as indirect measure for moist processes.

**Minor Comments and Typos:**

- L.8 What temporal frequency is this wave activity flux associated with? 10-day lowpass is it? Yes, we filter the final local wave activity by a 10-day low pass filter. We will make this clear in Sec. 2.3.2 and modified the text accordingly: (changes bold) "..filtered with the **10-day** low-frequency Lanczos filter introduced previously".

- L.22 typo, should be "investigates" Thank you.

- L. 38 it might be worth mentioning that it's not just forecast models that have trouble with blocking. GCM's do as well. Some references for this are Davini and D'Andrea (2020, Journal of Climate, From CMIP3 to CMIP6: NH Atmospheric Blocking...) and Narinesingh et al. (2022, Journal of Climate, Blocking in GFDL Models..) Thank you for this suggestion. We have included this point and the two suggested references at this point in the introduction.

- L. 56-58 This is a good place to cite Neal et al. (2022, GRL, The 2021 Pacific...) Thank you for this suggestion. We have included this reference.

- L. 95-96, how does this decomposition exhibit arbitrariness?
  The modified sentence now reads (sentence extended after 'arbitrariness'): "Arguably, this decomposition exhibits some degree of arbitrariness in the number of frequency bands of interest (Miller and Wang, 2022)".

- L. 96, define the temporal band of "low-frequency", i.e. 10-day lowpass? Yes, we added the specific information in the introduction: "..equation for low-frequency PV anomalies **(10-day low pass filtered)** that represent the evolution of blocked regimes.."

- L.144-145 Blocks are often considered to be phenomena that last at least 5 days (i.e. Dunn-Sigouin and Son 2013 algorithm). By doing a 10-day lowpass are you not filtering out blocks that last 5-10 days? The answer cannot simply be given in terms of yes or no. A blocking lasting less than 10 days can still be included in our investigation depending on the days before and after the block. In general only "true" waves with frequencies less than 10 days would be filtered out. However, a purely single positive or negative signal would remain even if it lasts less than 10 days. From our extensive experience with the year-round North Atlantic-European weather regimes, we do not see indication that relatively short regime life cycles were filtered by the 10-day lowpass filter. In fact, more than half of the blocked weather regime life cycles last 5-10 days.

- L. 149-150 Perhaps you can put the number of each events here or make a table listing the amount of each event to get a sense of how robust the statistics are. Following the suggestion of another reviewer we included the number of events contributing to our study in Figure 1.

- Figure 1. Caption, should be "year-round" not your-round. Thank you.

- L.197-199 I am confused about the timescales of the primes and how you are calculating anomalies. If you are doing 10-day lowpass filters won't this filter out the high-frequency eddy contributions from synoptic systems? Thank you for noting this potential source of confusion. We have revised the manuscript to provide a clearer explanation at this point. The revised version reads (last paragraph of subsection 2.3.1): "Note that primed variables in this study refer to deviations from a climatological background state. The EDDY term thus does not imply a decomposition into different frequency bands, as it often does in other studies. Equation 4 is derived by applying the low-frequency filter (denoted by subscript $L$) to the PV tendency equation, which

implies a low-frequency filter of the tendency terms on the RHS, but *not* of the individual variables involved in these terms. Further note that the tendency terms on the RHS of Equation 4 are deviations from their climatological averages. The derivation of Equation 4 is given in Appendix A, where we also explain why we do not consider nonconservative tendencies $\mathcal{N}$ explicitly in this study."

- L. 281-286 Again I am curious as to if the following features you go on to describe are the same for all seasons. We would like to refer to our answer of the reviewer's second comment, where we answer the seasonal aspects of our study in some detail.

**References**

Benedict, J. J., Lee, S., and Feldstein, S. B.: Synoptic View of the North Atlantic Oscillation, Journal of the Atmospheric Sciences, 61, 121–144, https://doi.org/10.1175/1520-0469(2004)061⟨0121:SVOTNA⟩2.0.CO;2, 2004.

Büeler, D., Ferranti, L., Magnusson, L., Quinting, J. F., and Grams, C. M.: Year-Round Sub-Seasonal Forecast Skill for Atlantic–European Weather Regimes, Quarterly Journal of the Royal Meteorological Society, 147, 4283–4309, https://doi.org/10.1002/qj.4178, 2021.

Caliński, T. and Harabasz, J.: A dendrite method for cluster analysis, Communications in Statistics-theory and Methods, 3, 1–27, 1974.

Feldstein, S. B.: The Dynamics of NAO Teleconnection Pattern Growth and Decay, Quarterly Journal of the Royal Meteorological Society, https://doi.org/10.1256/qj.02.76, 2003.

Ferranti, L., Magnusson, L., Vitart, F., and Richardson, D. S.: How far in advance can we predict changes in large-scale flow leading to severe cold conditions over Europe?, Quarterly Journal of the Royal Meteorological Society, 144, 1788–1802, 2018.

Hauser, S., Knippertz, P., Quinting, J. F., Riemer, M., Teubler, F., and Grams, C. M.: A process-based understanding of Greenland Blocking regime life cycle dynamics in ERA-5 reanalysis from a potential vorticity perspective, Tech. rep., Copernicus Meetings, https://doi.org/10.5194/ems2022-120, 2022a.

Hauser, S., Teubler, F., Riemer, M., Knippertz, P., and Grams, C. M.: Towards a Diagnostic Framework Unifying Different Perspectives on Blocking Dynamics: Insight into a Major Blocking in the North Atlantic-European Region, Weather and Climate Dynamics Discussions, pp. 1–36, https://doi.org/10.5194/wcd-2022-44, 2022b.

Hochman, A., Messori, G., Quinting, J. F., Pinto, J. G., and Grams, C. M.: Do Atlantic-European Weather Regimes Physically Exist?, Geophysical Research Letters, 48, e2021GL095 574, https://doi.org/10/gm527x, 2021.

Madonna, E., Li, C., Grams, C. M., and Woollings, T.: The Link between Eddy-Driven Jet Variability and Weather Regimes in the North Atlantic-European Sector, Quarterly Journal of the Royal Meteorological Society, 143, 2960–2972, https://doi.org/10/gjv4q9, 2017.

Martineau, P., Nakamura, H., Yamamoto, A., and Kosaka, Y.: Baroclinic Blocking, Geophysical Research Letters, 49, e2022GL097 791, https://doi.org/10.1029/2022GL097791, 2022.

Michel, C. and Rivière, G.: The Link between Rossby Wave Breakings and Weather Regime Transitions, Journal of the Atmospheric Sciences, 68, 1730–1748, https://doi.org/10.1175/2011JAS3635.1, 2011.

Michel, C., Rivière, G., Terray, L., and Joly, B.: The Dynamical Link between Surface Cyclones, Upper-Tropospheric Rossby Wave Breaking and the Life Cycle of the Scandinavian Blocking, Geophysical Research Letters, 39, https://doi.org/10.1029/2012GL051682, 2012.

Nakamura, N. and Huang, C. S. Y.: Atmospheric Blocking as a Traffic Jam in the Jet Stream, Science, 361, 42–47, https://doi.org/10.1126/science.aat0721, 2018.

Teubler, F. and Riemer, M.: Dynamics of Rossby Wave Packets in a Quantitative Potential Vorticity–Potential Temperature Framework, Journal of the Atmospheric Sciences, 73, 1063–1081, https://doi.org/10.1175/JAS-D-15-0162.1, 2016.

Teubler, F. and Riemer, M.: Potential Vorticity Dynamics of Troughs and Ridges within Rossby Wave Packets during a 40-Year Reanalysis Period, Weather and Climate Dynamics Discussions, pp. 1–24, 2021.

Thorndike, R. L.: Who belongs in the family, in: Psychometrika, Citeseer, 1953.

Wang, L. and Kuang, Z.: Atmospheric blocking as an evolution of Rossby wave packets, in: AGU Fall Meeting Abstracts, vol. 2019, pp. A43B–04, 2019.

Wirth, V., Riemer, M., Chang, E. K. M., and Martius, O.: Rossby Wave Packets on the Midlatitude Waveguide—A Review, Monthly Weather Review, 146, 1965–2001, https://doi.org/10.1175/MWR-D-16-0483.1, 2018.

Yeh, T.-c.: On Energy Dispersion in the Atmosphere, Journal of the Atmospheric Sciences, 6, 1–16, https://doi.org/10.1175/1520-0469(1949)006⟨0001:OEDITA⟩2.0.CO;2, 1949.

---

## Author Response (AR2)

**Response to Reviewers**

Franziska Teubler, Michael Riemer, Christopher Polster,
Christian M. Grams, Seraphine Hauser, and Volkmar Wirth

March 15, 2023

**Contents**

**1 Response to Gwendal Rivière**

The authors have deeply and adequately considered my questions and suggestions. I am very satisfied with the revised version of the paper. Congratulations! Before submitting the final version, I would encourage the authors to check consistencies in the notations.

Thank you very much for your detailed and very thorough review. We appreciate the time and effort a lot you put into the review.

- Line 170 $\Phi^L$, hence variable $\Phi$ with the superscript L, is said to be the low-frequency geopotential height anomaly. But line 191, $q'_L$, hence with the subscript L and the prime, is used to denote the low-frequency PV anomaly. Line 253, $q'^L$, hence with the superscript L, is used to denote the same quantity. It is important to have a homogeneous notation throughout the paper to denote the low-frequency anomaly. One possibility is to use the operator L when both the low-pass filter is applied and the climatological mean is subtracted, this will avoid to have primes and L at the same time. For variables for which only the climatological mean is subtracted, you could use the prime as it is already the case. Indeed this is inconsistent. We added a prime for the geopotential height anomaly in L170 to make this clear and decided to use everywhere the subscript L.

- Section 2.3.2: the prime appearing in that section does not mean the same thing as in section 2.3.1. Please find another notation for that section. Maybe a hat ? Again, good point. We even mentioned that the prime is associated with deviations from a climatological background state and hence followed your recommendation to use a hat in Section 2.3.2. to avoid confusion.

**2 Response to Reviewer 2 and 3**

As well, we would like to thank Review 2 and 3 for their time, effort and ideas to improve our manuscript.